# Air Quality Modeling with the Use of Regression Neural Networks

**DOI:** 10.3390/ijerph192416494

**Published:** 2022-12-08

**Authors:** Szymon Hoffman, Mariusz Filak, Rafał Jasiński

**Affiliations:** Faculty of Infrastructure and Environment, Czestochowa University of Technology, 69 Dabrowskiego St., 42-200 Czestochowa, Poland

**Keywords:** air quality, air monitoring, air pollutants, regression, prediction, prediction error, artificial neural networks

## Abstract

Air quality is assessed on the basis of air monitoring data. Monitoring data are often not complete enough to carry out an air quality assessment. To fill the measurement gaps, predictive models can be used, which enable the approximation of missing data. Prediction models use historical data and relationships between measured variables, including air pollutant concentrations and meteorological factors. The known predictive air quality models are not accurate, so it is important to look for models that give a lower approximation error. The use of artificial neural networks reduces the prediction error compared to classical regression methods. In previous studies, a single regression model over the entire concentration range was used to approximate the concentrations of a selected pollutant. In this study, it was assumed that not a single model, but a group of models, could be used for the prediction. In this approach, each model from the group was dedicated to a different sub-range of the concentration of the modeled pollutant. The aim of the analysis was to check whether this approach would improve the quality of modeling. A long-term data set recorded at two air monitoring stations in Poland was used in the examination. Hourly data of basic air pollutants and meteorological parameters were used to create predictive regression models. The prediction errors for the sub-range models were compared with the corresponding errors calculated for one full-range regression model. It was found that the application of sub-range models reduced the modeling error of basic air pollutants.

## 1. Introduction

Air pollution is considered one of the main factors affecting the human population and the environment. The World Health Organization estimates that many millions of people die prematurely due to poor air quality [1]. Once pollutants are emitted into the air, it is impossible to stop them. If pollutants get into the atmosphere, they contribute to the deterioration of air quality in the vicinity of the emission, but by spreading they can have negative effects hundreds and thousands of kilometers from the point of emission. Therefore, air pollution is treated as a global threat and emission reduction strategies are implemented in many countries. It should be highlighted that, in Poland, the PM_10_ (particulate matter) concentration still exceeds the permissible limits and causes premature deaths of over 40,000 people every year [2].

Air pollution causes negative changes in the human respiratory and circulatory systems, even when pollutant concentrations do not exceed permissible levels [3,4,5,6,7,8]. It may trigger various reactions of organisms, including mental health disorders [9,10]. It was also found that air pollution can negatively affect the economy [11,12,13,14]. High concentrations of air pollutants have a negative effect on plants, which is manifested in the reduction of crop yields in agriculture [15,16].

The control and reduction of anthropogenic emissions are now recognized as the keys to good global air quality. An important element of the control system is the assessment of air quality. This task is performed using air quality monitoring. Air monitoring includes continuous measurements of air pollutants. The main air pollutants include O_3_, SO_2_, NO_x_, CO and PM_10_ [17]. Concentrations of the mentioned pollutants can be measured with automatic air monitoring stations. These types of stations are often equipped with devices that also continuously measure meteorological data such as temperature, wind direction, wind speed and solar radiation. Measurement results at automatic air monitoring stations are recorded in the form of 1 h averages, e.g., hourly concentrations, hourly temperature, etc. In the EU, average hourly concentrations are the basis for calculating averages over longer periods of time that are required for air quality standards, such as 8 h, 24 h and annual concentrations [18]. The system acquires and collects measurement data of the air pollutants’ concentration levels at many individual air monitoring stations, according to the standardized measurement methods [19,20,21,22,23]. The collected hourly concentrations in the air monitoring system constitute the basis for direct and indirect statistical evaluation of air quality in the zones represented with individual monitoring stations, in accordance with the procedures described in the relevant legal acts [18]. Correct assessment requires a high degree of completeness of the time series of concentrations obtained at the monitoring stations. Usually, the completeness should exceed 90% in an annual series of hourly measurements. Unfortunately, monitoring data are never 100% complete in annual terms, and often they do not even have the completeness required by the regulations. When there is a deficit of data in a series of measurements, then there is a need to complete the missing data.

Missing data can be supplemented by introducing modeled concentrations in the measurement gaps [24,25,26,27]. The time series data obtained with air monitoring have specific characteristics. All measurements, both of concentrations and meteorological parameters, are performed simultaneously and are recorded in similar time series. Therefore, methods based on auto-regression and regression can be used for modeling monitoring concentrations. If historical data from the selected air monitoring station are available, they can be used to explore the knowledge hidden in them. Autonomous models of this type can be accurate and have a very significant advantage: the approximation of concentrations does not require external data from outside the monitoring system [26,27]. In the first models, classical statistical regression techniques were used [28]. Classical methods are increasingly being replaced by methods that use machine learning artificial intelligence, including artificial neural networks (ANNs) [29,30,31,32,33,34,35,36,37]. ANN models enable a deeper exploration of knowledge hidden in historical data and, as a result, more accurate concentration predictions.

In regression modeling, it was found that the application of one neural network to the entire range of concentrations of the predicted pollutant resulted in different prediction accuracies in the concentration sub-ranges [38,39]. It was considered advisable to replace one neural network with several networks (sub-models), each of which would be adjusted to specific concentration sub-ranges [39]. The use of several sub-range models should improve the accuracy of the prediction. This paper presents an analysis that verifies the above thesis.

The main aim of the study was to improve the accuracy of prediction of air pollutant concentrations in neural regression models, using many predictive models created for various sub-ranges of air pollutant concentrations. The analyzed data came from two air monitoring stations in the Upper Silesian Region, Poland. The quality of prediction was assessed separately for the concentrations of six main air pollutants: O_3_, NO, NO_2_, SO_2_, PM_10_ and CO. Multi-layer perceptrons with an identical architecture were used to model all pollutants. The predicted concentrations were compared with the observed ones to estimate the prediction error. The prediction errors were calculated for various sub-models, and, based on them, the prediction error was estimated over the entire concentration range. This error was compared with the approximation error obtained from a single model covering the entire concentration range. 

## 2. Materials and Methods

### 2.1. Air Monitoring Data

Hourly data from two air monitoring stations located in a highly industrialized region of Poland were used for the analysis. The data were recorded in Zabrze and Złoty Potok in the years 2011–2016. The location of selected monitoring stations is shown in Figure 1. 

The city of Zabrze is one of the most polluted towns in Poland and throughout the EU. The station at Zabrze was an urban background monitoring site. Złoty Potok is a rural town, located outside the Upper Silesian Agglomeration. In Złoty Potok, there is a background monitoring station for the Upper Silesian Region. The data were provided by the Voivodeship Inspectorate of Environmental Protection in Katowice. Time series data, including 1 h average values of O_3_, NO, NO_2_, SO_2_, PM_10_ and CO concentrations, as well as meteorological data for temperature, wind speed, solar radiation and relative humidity, were recorded. The analyzed data sets are not publicly available. We received them on an individual request from the Voivodeship Inspectorate for Environmental Protection. Time series of concentrations are available on the website of the Chief Inspectorate of Environmental Protection: https://powietrze.gios.gov.pl/pjp/archives, accessed on 30 April 2022. This is only part of the data. Meteorological data are not available online. The data also include two variables describing the time: day and hour. These two variables were converted to numeric form following the procedure described in [39].

The following symbols were used to describe the time series (variables):

D—numeric day,

H—numeric hour,

O_3_—hourly average of O_3_ concentration (µg/m^3^),

NO—hourly average of NO concentration (µg/m^3^),

NO_2_—hourly average of NO_2_ concentration (µg/m^3^),

SO_2_—hourly average of SO_2_ concentration (µg/m^3^),

CO—hourly average of CO concentration (mg/m^3^),

PM_10_—hourly average of PM_10_ concentration (µg/m^3^),

WS—hourly average of wind speed (m/s),

T—hourly average of air temperature (°C),

I—hourly average of solar radiation intensity (W/m^2^),

RH—hourly average of relative humidity (%).

Descriptive statistics of the 6-year set of hourly concentrations of monitored air pollutants are presented in Table 1.

### 2.2. Regression Models

Multi-layer perceptrons with identical architecture were used to model all pollutants. The predicted concentrations were compared with the observed ones to estimate the prediction error. The prediction errors were calculated for various sub-models, and, based on them, the prediction error was estimated over the entire concentration range. This error was compared with the approximation error obtained from a single model covering the entire concentration range. For all air pollutants, similar perceptron models were created. The output of this was the concentration of the chosen air pollutant (explained variable), and the inputs were the date and hour, concentrations of other air pollutants and meteorological parameters (explanatory variables). Table 2 presents the predicted variables and predictors, with separate lines for individual models. For example, the following 11 input variables were used to model the ozone concentration in Złoty Potok: H, D, NO, NO_2_, SO_2_, CO, PM_10_, WS, T, I and RH.

For regression modeling, artificial neural networks in the form of multi-layer perceptrons were used. Each perceptron consisted of input neurons, with 10 neurons in one hidden layer and one output neuron. Figure 2 shows such a perceptron used to model O_3_ concentrations. The Broyden–Fletcher–Goldfarb-Shanno (BFGS) algorithm was used in the learning process. The Broyden–Fletcher–Goldfarb–Shanno algorithm is used in problems related to numerical optimization [40]. The BFGS algorithm was developed on the basis of solutions proposed in 1970 by the four mathematicians mentioned in its name [41,42,43,44]. The algorithm uses an iterative method of solving unlimited non-linear optimization problems.

The learning process was always limited to 300 epochs. An epoch is a specialized expression related to the learning process of a neural network. In the network learning process, one epoch means a single learning cycle. In the learning process, the network repeats the cycles many times in order to minimize the learning error. The function of activating hidden and output neurons was a logistic function. The scales were initiated randomly. The network initialization was random (Gaussian). Modeling was performed using the Statistica program, version 13.3. Each prediction was performed 5 times. Repeating the training of the neural network, while maintaining the same parameters of the learning process, is a routine procedure. Each model’s learning process is somewhat random and leads to a different network. The created networks have the same structure of neurons, but differ in terms of weights and the degree of activation of individual neurons. In general, they differ slightly in the modeling error. The most accurate of the 5 created models was selected for reporting. The other models were rejected. The sum of squares (SOS) was assumed as the error function. SOS is the sum of the squared distances of all predicted values from the actual values.

### 2.3. Preparation of Data for Modeling

The complete set of hourly air monitoring data from the 6-year measurement period should have included 52,608 cases (hourly observations). Prior to modeling, raw data from the air monitoring database were prepared by removing the cases where there were missing data. After removing the cases with missing data, a set of cases with complete data was obtained for further analysis. This set was called the full-range set. The full-range sets included 36,460 and 15,536 cases, for Zabrze and Złoty Potok stations, respectively. Before starting the learning process, each data set was divided into three subsets: the training subset consisted of 70% of the cases, the validation subset consisted of 15% of the cases and the test subset consisted of 15% of the cases. 

Two approaches were adopted during modeling and, therefore, two groups of models were developed for each of the stations. In the first approach, all cases in the data set were sorted according to the real value of the predicted variable, from the lowest concentration to the highest. Then, a set of cases sorted in this way was divided into subsets. The entire set of cases was divided gradually into 2, then 4, and finally 8 subsets of the same size. The subsets prepared in this way differed in the ranges of the real concentration values of the predicted variable. For each of these subsets, regression models called RVS (Real Values Sorting) were created. As a result, for each set/subset, a separate predictive model was obtained, marked with the same symbol as the modeled set/subset. The name of the monitoring station (ZAB or ZP) was also added to the designations of sets and subsets. The full-range model was marked as RVS-1/1. As a result of the division of the full-range set, the following subsets and submodels were obtained:− Two sub-models (RVS-1/2, RVS-2/2), after division into two subsets;− Four sub-models (RVS-1/4, RVS-2/4, RVS-3/4, RVS-4/4), after division into four subsets;− Eight sub-models (RVS-1/8, RVS-2/8, RVS-3/8, RVS-4/8, RVS-5/8, RVS-6/8, RVS-7/8, RVS-8/8), after division into eight subsets. 

In a situation where the real concentrations of the pollutant to be modeled were not known, the RVS-type models could not be used because it was not possible to sort and classify the cases into the real concentration sub-ranges. Therefore, RVS models reflect only the potential, not practical, opportunities to improve the quality of modeling through segmentation of the prediction process. If the real concentrations are not known and there is a need to predict them, then the RVS sub-range models are not available. In such a situation, a different approach can be proposed, which also uses division into sub-ranges, but then with sectoral modeling in designated sub-ranges. The most important step in this approach is the initial modeling of the concentrations of the selected pollutant in the full-range set, understood as a set of all cases containing complete data of all explanatory variables (model inputs) for the modeled pollutant. After initial modeling of the entire range of cases, predictive concentrations of the dependent variable are obtained. Then, the entire set of cases is sorted according to the increasing predictive concentrations of the modeled pollutant. The next step is to divide the sorted full-range set of cases into a specified number of equal sub-ranges. In this way, you can generate a model called PVS (Predicted Values Sorting). As a result of the division of the full-range set, the following subsets and submodels were obtained:− Two sub-models (PVS-1/2, PVS-2/2), after division into two subsets;− Four sub-models (PVS-1/4, PVS-2/4, PVS-3/4, PVS-4/4), after division into four subsets;− Eight sub-models (PVS-1/8, PVS-2/8, PVS-3/8, PVS-4/8, PVS-5/8, PVS-6/8, PVS-7/8, PVS-8/8), after division into eight subsets. 

The scheme of dividing the full-range set into sub-ranges is shown in Figure 3 and Figure 4 for both monitoring stations.

Thanks to the divisions of the full-range set, it was possible to check how the modeling accuracy changed in the concentration sub-ranges, and whether the modeling carried out in the sub-ranges would improve the modeling quality in relation to the full-range modeling.

### 2.4. Assessment of the Approximation Error

To assess the accuracy of the obtained regression models, the MAE and RMSE values were used, which were calculated on the basis of the discrepancy between the actual and predicted values. The formulas for calculating individual errors are presented in Equations (1) and (2).

*MAE*—mean absolute error
(1)$$MAE=\frac{1}{n}\overset{n}{\underset{i=1}{\sum }}\left|x_{i}-y_{i}\right|$$

*RMSE*—root-mean-squared error
(2)$$RMSE=\sqrt{\frac{\sum_{i=1}^{n}{\left(x_{i}-y_{i}\right)}^{2}}{n}}$$
where *n*—number of cases, *y*—predicted concentrations, *x*—real concentrations, *i*—the case number.

## 3. Results

For each pollutant, modeling errors were calculated in relation to the real pollutant concentrations (O_3_, NO, NO_2_, SO_2_, CO and PM_10_ for the Zabrze station; and O_3_, NO, NO_2_, SO_2_ and PM_10_ for the Zloty Potok station). To assess the modeling accuracy, two error measures were calculated: MAE and RMSE. 

### 3.1. The Results of the Modeling of O_3_ Concentrations

Table 3, Table 4, Table 5 and Table 6 show the results of O_3_ concentration predictions obtained with the full-range and sub-range models. The errors of the PVS models for the Zabrze monitoring station are presented in Table 3 and the Złoty Potok monitoring station in Table 4. Similar lists of errors for the RVS models are presented in Table 5 and Table 6. The presented results show that the sub-range modeling errors changed for individual sub-ranges. Regardless of the number of sub-range models, the modeling error usually increased with increasing concentration values in the sub-ranges. A good way to assess the quality of the prediction is to compare the overall prediction error over the entire range of concentrations of the modeled pollutant, and not in individual sub-ranges. Therefore, the tables include average values for the entire ranges: “overall MAE” and “overall RMSE”.

Division into the sub-ranges generally improved the accuracy of prediction, especially in the case of RVS models. The exceptions were the eight-sub-range PVS models for both monitoring stations. In the case of Zabrze, the same average value of MAE was recorded in the eight-sub-range models as in the four-sub-range models (8.02 µg/m^3^). In the case of Zloty Potok, even an increase in the overall values of MAE and RMSE errors was observed in the eight-sub-range models compared to the overall error of the four-sub-range models. In the case of RVS models, a decrease in the overall values of MAE and RMSE was always observed with an increase in the number of sub-ranges. The comparison showed that dividing the area into sub-ranges and separately modeling these sub-ranges improved the overall quality of the prediction, but that having too many sub-ranges and sub-models may be ineffective.

The values of “overall MAE” and “overall RMSE” depending on the number of sub-ranges are presented in Figure 5, Figure 6, Figure 7 and Figure 8. 

### 3.2. The Results of the Modeling of NO Concentrations

Table 7, Table 8, Table 9 and Table 10 show the results of NO concentration predictions obtained with the full-range and sub-range models. The errors of the PVS models for the Zabrze monitoring station are presented in Table 7 and the Złoty Potok monitoring station in Table 8. The corresponding errors for the RVS models are presented in Table 9 and Table 10. The division into sub-ranges generally improved the accuracy of prediction, especially in the case of RVS models. In general, as the number of sub-models increased, the overall measures of modeling error decreased. The exceptions were the eight-sub-range PVS models for the monitoring station at Złoty Potok. At this station, the overall MAE value for the eight-sub-range sub-models (0.429 µg/m^3^) was comparable to the overall MAE value for the four-sub-range sub-models (0.428 µg/m^3^). In the case of RVS models, a decrease in the overall values of MAE and RMSE errors was always observed with an increase in the number of sub-ranges. In the Zloty Potok station, for the real concentrations in sub-ranges with a range of 0.0-0.0 µg/m^3^ and 1.0-1.0 µg/m^3^, RVS sub-models could not be created due to the lack of variability in these concentration ranges. Therefore, the overall error for this group of sub-models was not estimated. Figure 9, Figure 10, Figure 11 and Figure 12 show the overall values of MAE and RMSE graphically.

### 3.3. The Results of the Modeling of NO_2_ Concentrations

Table 11, Table 12, Table 13 and Table 14 show the results of NO_2_ concentration predictions obtained with the full-range and sub-range models. The errors of the PVS models for the Zabrze monitoring station are presented in Table 11 and the Złoty Potok monitoring station in Table 12. The corresponding errors for the RVS models are presented in Table 13 and Table 14. The division into sub-ranges generally improved the accuracy of prediction, especially in the case of RVS models. In general, as the number of sub-models increased, the overall measures of modeling error decreased. The exceptions were the eight-sub-range PVS models for both monitoring stations. The overall MAE and RMSE values for the eight-sub-range sub-models were higher than the overall MAE and RMSE values for the four-sub-range sub-models. In the case of RVS models, a decrease in the overall values of MAE and RMSE errors was always observed with an increase in the number of sub-ranges. In the case of PVS models, it appeared that having too many sub-ranges and sub-models could degrade the quality of the modeling. Figure 13, Figure 14, Figure 15 and Figure 16 show the overall values of MAE and RMSE graphically.

### 3.4. The Results of the Modeling of SO_2_ Concentrations

Table 15, Table 16, Table 17 and Table 18 show the results of SO_2_ concentration predictions obtained with the full-range and sub-range models. The errors of the PVS models for the Zabrze monitoring station are presented in Table 15 and the Złoty Potok monitoring station in Table 16. The corresponding errors for the RVS models are presented in Table 17 and Table 18. The division into sub-ranges generally improved the accuracy of prediction, especially in the case of RVS models. In general, as the number of sub-models increased, the overall measures of modeling error decreased. The exceptions were the four-sub-range and eight-sub-range PVS sub-models for the monitoring station at Zabrze.

At this station, the overall MAE values (5.18 µg/m^3^ for the eight-sub-range sub-models, and 5.17 µg/m^3^ for the four-sub-range sub-models) were comparable to the overall MAE value for the two-sub-range sub-models (5.17 µg/m^3^). In the case of RVS models, a decrease in the overall values of MAE and RMSE was always observed with an increase in the number of sub-ranges. In the case of PVS models, it appeared that having too many sub-ranges and sub-models could degrade the quality of the modeling. Figure 17, Figure 18, Figure 19 and Figure 20 show the overall values of MAE and RMSE graphically.

### 3.5. The Results of the Modeling of PM_10_ Concentrations

Table 19, Table 20, Table 21 and Table 22 show the results of PM_10_ concentration predictions obtained in the full-range and sub-range models. The errors of the PVS models for the Zabrze monitoring station are presented in Table 19 and the Złoty Potok monitoring station in Table 20. The corresponding errors for the RVS models are presented in Table 21 and Table 22. The division into sub-ranges generally improved the accuracy of prediction, especially in the case of RVS models. In general, as the number of sub-models increased, the overall measures of modeling error decreased. The exceptions were the eight-sub-range PVS models for the monitoring station at Złoty Potok. The overall MAE and RMSE values for the eight-sub-range models were higher than the overall MAE and RMSE values for the four-sub-range models and even the two-sub-range models. In the case of RVS models, a decrease in the overall values of MAE and RMSE was always observed with an increase in the number of sub-ranges. In the case of PVS models, it appeared that having too many sub-ranges and submodels could degrade the quality of the modeling. Figure 21, Figure 22, Figure 23 and Figure 24 show the overall values of MAE and RMSE graphically.

### 3.6. The Results of the Modeling of CO Concentrations

CO concentrations were not monitored at the Złoty Potok station, so the analysis was carried out only using monitoring data from Zabrze. 

Table 23 and Table 24 show the results of PM_10_ concentration predictions obtained with the full-range and sub-range models. The errors in the PVS models for the Zabrze monitoring station are presented in Table 23. The errors in the RVS models are presented in Table 24. The division into sub-ranges improved the accuracy of prediction in the case of RVS models. In general, as the number of sub-models increased, the overall measures of modeling error decreased. The PVS models showed slight changes in accuracy. The MAE level did not change much. The overall MAE value for the eight-sub-range models was higher than the overall MAE value for the four-sub-range models and equal to the MAE value for the two-sub-range models. In the case of RVS models, a decrease in the overall values of MAE and RMSE was always observed with an increase in the number of sub-ranges. In the case of PVS models, it appeared that having too many sub-ranges and sub-models could degrade the quality of the modeling. Figure 25 and Figure 26 show the overall values of MAE and RMSE graphically.

## 4. Summary and Discussion

Table 25 shows the percentage changes in the overall values of MAE and RMSE obtained by modeling the concentrations in sub-ranges, calculated in relation to the error values of the corresponding full-range models. 

In the case of RVS models, each division into narrower concentration sub-ranges and the development of appropriate sub-range models resulted in a significant reduction in the overall value of the modeling error. The division into sub-models always improved the accuracy of predictions in the case of RVS models. When the number of sub-models increased, the overall measures of modeling error decreased. A significant improvement in the quality of modeling was achieved at both air monitoring stations. Modeling errors could be reduced by more than 60% using eight sub-models. However, it should be emphasized that the RVS models are not of great practical importance because their use is related to knowledge of the real concentrations of the pollutants. Moreover, once the concentration values are known, there is no need to perform modeling. The importance of the RVS models was that they allowed us to assess the potential for improving the quality of the modeling.

The division into sub-models generally improved the accuracy of the PVS models; however, the decrease in modeling error was not as great as in the RVS models. Moreover, quite often, after splitting the full-range set into eight sub-ranges and running eight sub-models, the MAE and RMSE values could be higher than in the sub-models created after division into only four sub-ranges. Such an effect was found for O_3_, NO, NO_2_ and PM_10_ in Złoty Potok, and NO_2_, SO_2_ and CO in Zabrze. The probable cause of the deterioration in the quality of prediction in some eight sub-range PVS models was the error in the classification of cases into individual sub-ranges. The classification was made on the basis of the predicted concentration values obtained as a result of the preliminary prediction. With an increasing number of sub-ranges, the sub-ranges of concentrations became narrower and the number of misclassified cases also increased. The number of misclassified cases became so large that it increased the mean prediction error in the sub-range. 

The PVS models always showed a lower accuracy than the RVS models. This is understandable, as the PVS models required a preliminary prediction, which introduced an additional error. In conclusion, having too many sub-ranges and sub-models can degrade the quality of modeling for the PVS models.

After division into sub-ranges (two sub-ranges, four sub-ranges, eight sub-ranges), the error in the models for the highest sub-ranges was always very large compared to the error in the models for the lower sub-ranges. There are two reasons for this effect. The first is that the width of the highest sub-range was always greater than that of the models of the lower sub-ranges. The second reason was the need to predict extremely high concentrations. The approximation of such unusual concentrations is always burdened with a higher error.

## 5. Conclusions

Monitoring data are often not complete enough to carry out an air quality assessment. To fill the measurement gaps, predictive models are used. Such models often use archival measurement data from air monitoring systems. This is the best source of knowledge about the relationships between measured variables (concentrations and meteorological parameters). There is a need to model the missing concentrations as accurately as possible. The use of artificial neural networks reduces the prediction error compared to classical regression methods. In previous studies, a single regression model over the entire concentration range was used to approximate the concentrations of a selected pollutant. In this study, it was assumed that not a single model, but a group of models, could be used for the prediction. In this approach, each model from the group is dedicated to a different sub-range of the concentration of a modeled pollutant. The aim of this analysis was to check whether this approach would improve the quality of modeling. 

The aim of the analysis was not to create the most up-to-date models based on possible new data. Once trained, a model using historical data (e.g., from 2011 to 2016) should also be able to predict concentrations for current data. This feature of the model’s “generalization of acquired knowledge” was tested during the learning process on the cases from the test subset, and also on the cases from the validation subset after the network training was completed. The performed validation should show that the model can approximate the target value using data independent of the learning process. 

Air monitoring data from the period 2011–2016 allowed us to verify the possibility of improving the accuracy of modeling by carrying out modeling in subsets. A similar analysis can be carried out using data from other monitoring stations, or data from Zabrze and Złoty Potok stations from a different period. However, the selected set of cases should cover a measurement period of several years, so that the recorded cases correspond to different meteorological situations and different ranges of concentrations of monitored pollutants.

The most important conclusions that resulted from the conducted analysis are as follows:

Modeling segmentation, consisting in prediction in the sub-ranges of concentrations of the modeled pollutant, allowed for a higher overall modeling accuracy.For RVS models, segmentation of the modeling process guarantees a significant increase in modeling accuracy compared to a model based on the full-range of concentrations.In the case of PVS models, segmentation of the modeling process allows to reduce overall prediction errors. However, the number of concentration sub-ranges cannot be too large. When predicting with 8 submodels, the modeling accuracy may be lower than when predicting with 4 submodels.

The authors are aware that the article has its shortcomings and limitations. The analysis was carried out using historical data from only two air monitoring stations. Stations from the same region of Poland were selected. We are convinced that a similar analysis of completely different data, e.g., from other regions of Poland or other countries/continents, would confirm the formulated conclusions. It can be assumed with high probability that the concept is valid and should yield similar results for other air monitoring data sets. The effect was confirmed for six air pollutants. Not all possible air pollutants, e.g., PM_2.5_, were included. The authors are convinced that a similar effect could be obtained for other monitored pollutants.

## Figures and Tables

**Figure 1 ijerph-19-16494-f001:**
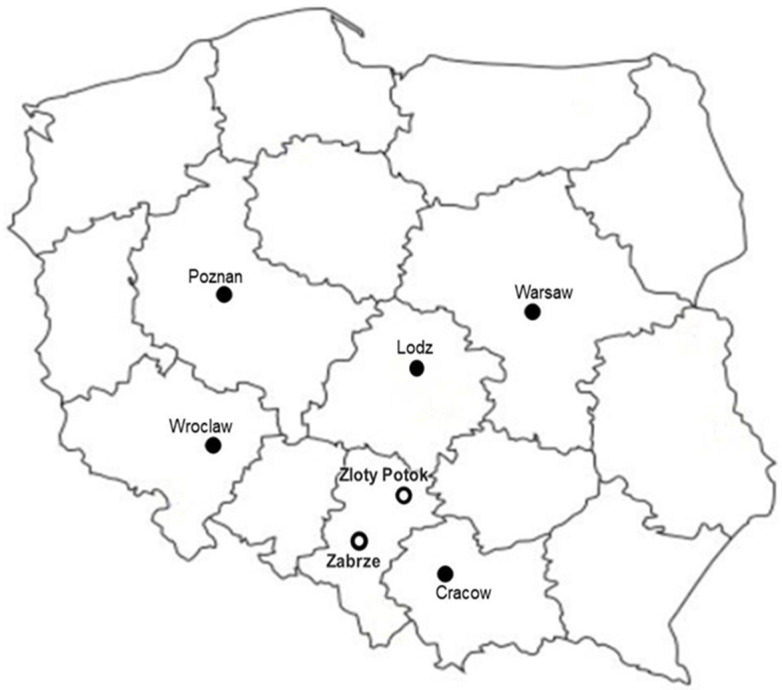
Locations of the air monitoring stations Zabrze and Złoty Potok in Poland.

**Figure 2 ijerph-19-16494-f002:**
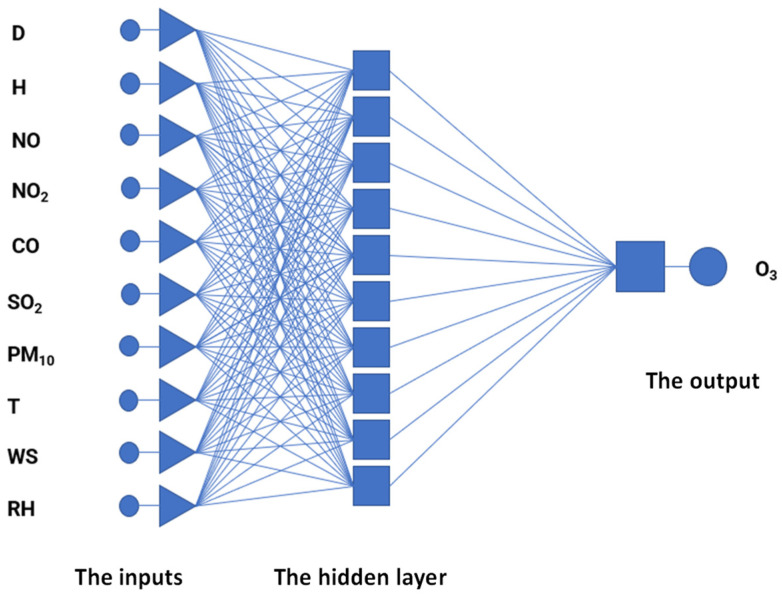
An architecture diagram of the multi-layer perceptron with ten neurons in a single hidden layer.

**Figure 3 ijerph-19-16494-f003:**
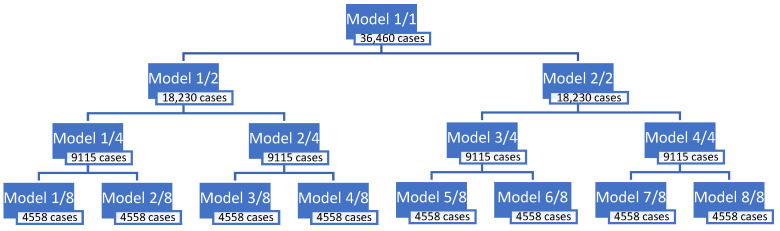
The scheme of the division into sub-ranges for the Zabrze station.

**Figure 4 ijerph-19-16494-f004:**
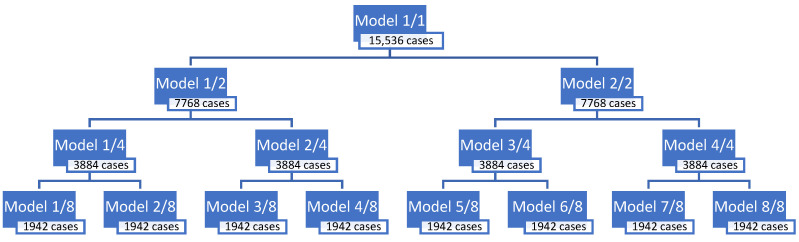
The scheme of the division into sub-ranges for the Złoty Potok station.

**Figure 5 ijerph-19-16494-f005:**
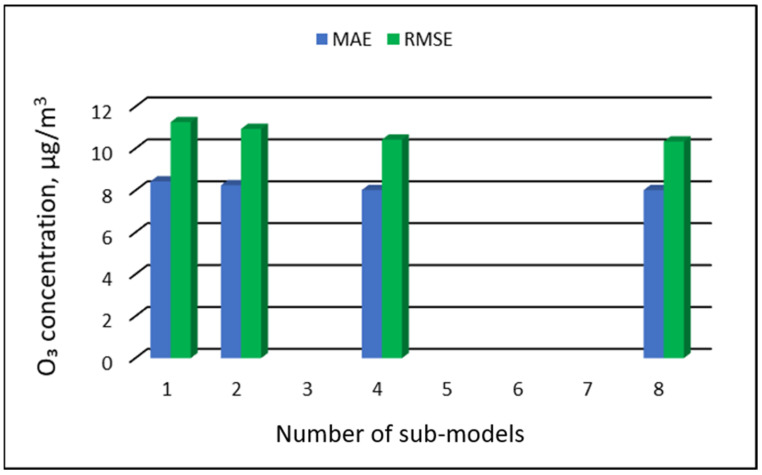
Overall MAE and RMSE values for O_3_ concentration prediction in PVS models depending on the number of created sub-models, Zabrze.

**Figure 6 ijerph-19-16494-f006:**
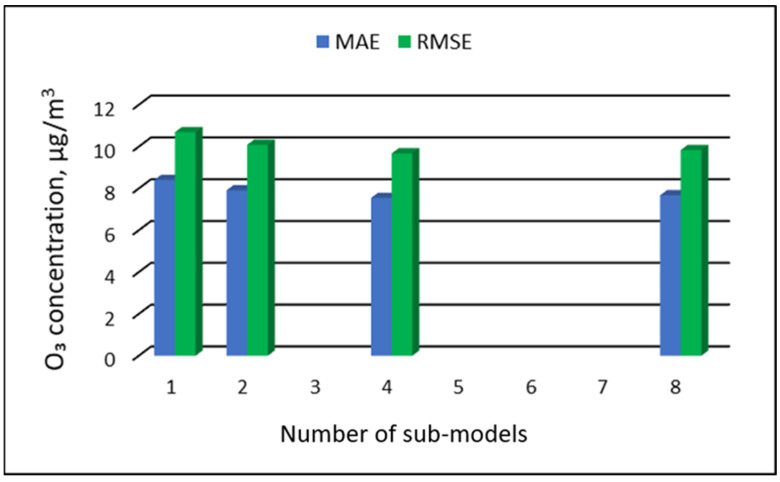
Overall MAE and RMSE values for O_3_ concentration prediction in PVS models depending on the number of created sub-models, Złoty Potok.

**Figure 7 ijerph-19-16494-f007:**
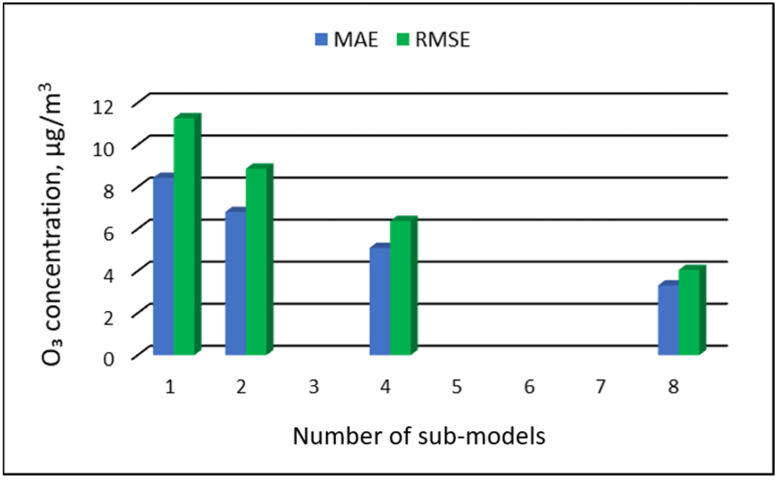
Overall MAE and RMSE values for O_3_ concentration prediction in RVS models depending on the number of created sub-models, Zabrze.

**Figure 8 ijerph-19-16494-f008:**
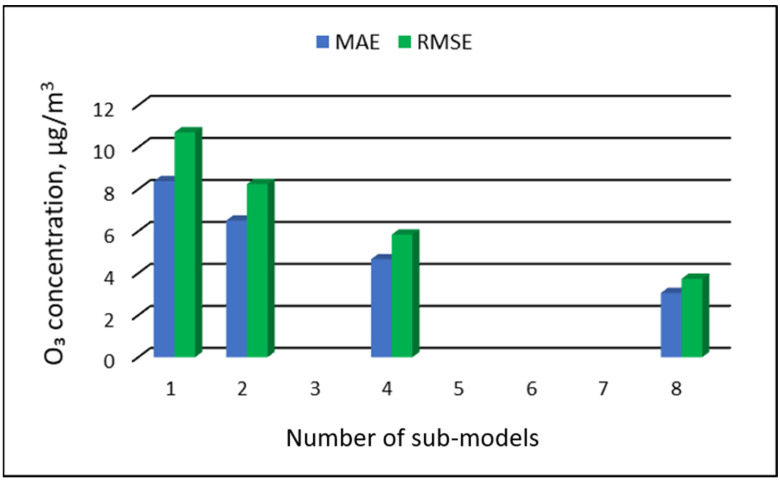
Overall MAE and RMSE values for O_3_ concentration prediction in RVS models depending on the number of created sub-models, Złoty Potok.

**Figure 9 ijerph-19-16494-f009:**
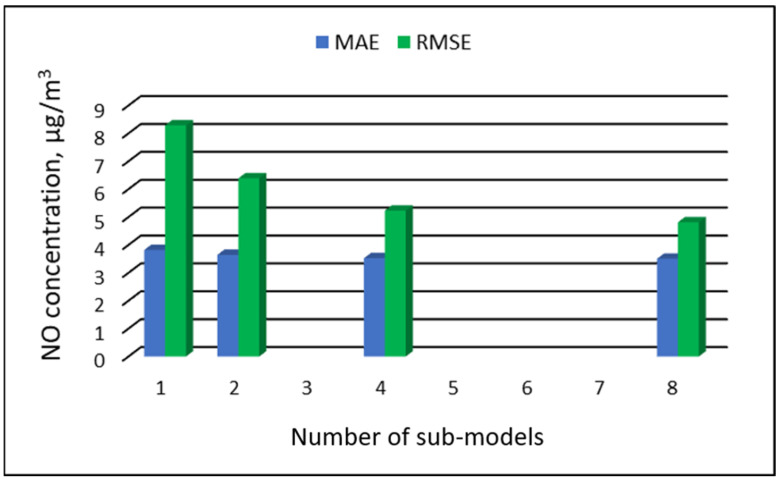
Overall MAE and RMSE values for NO concentration prediction in PVS models depending on the number of created sub-models, Zabrze.

**Figure 10 ijerph-19-16494-f010:**
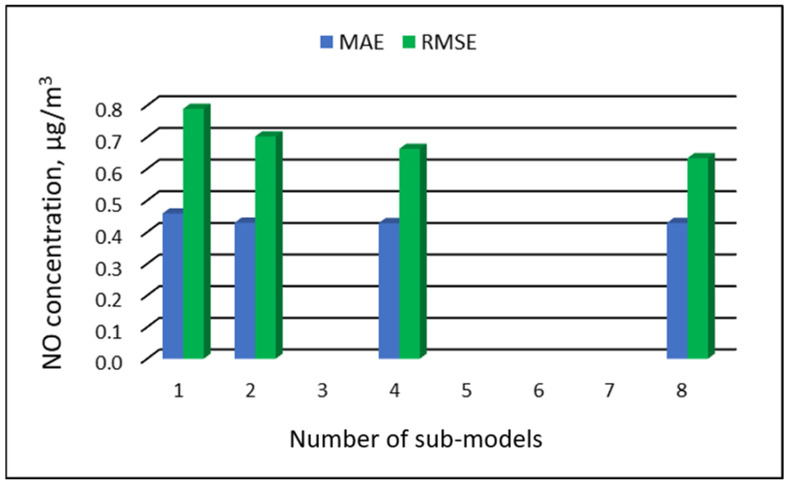
Overall MAE and RMSE values for NO concentration prediction in PVS models depending on the number of created sub-models, Złoty Potok.

**Figure 11 ijerph-19-16494-f011:**
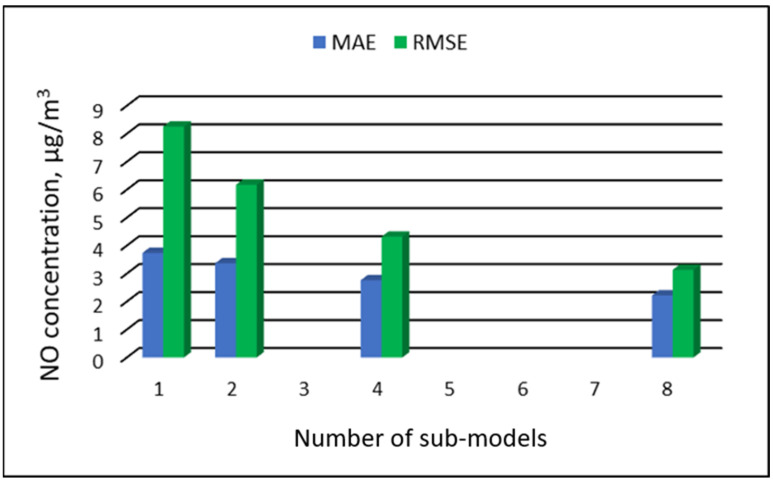
Overall MAE and RMSE values for NO concentration prediction in RVS models depending on the number of created sub-models, Zabrze.

**Figure 12 ijerph-19-16494-f012:**
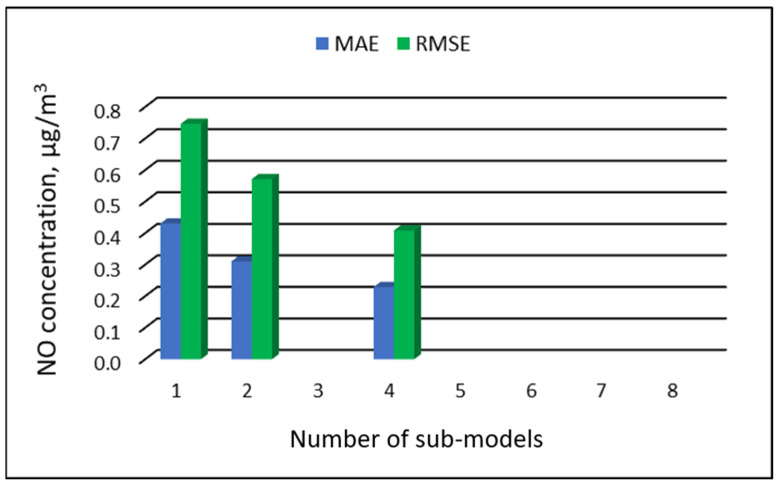
Overall MAE and RMSE values for NO concentration prediction in RVS models depending on the number of created sub-models, Złoty Potok.

**Figure 13 ijerph-19-16494-f013:**
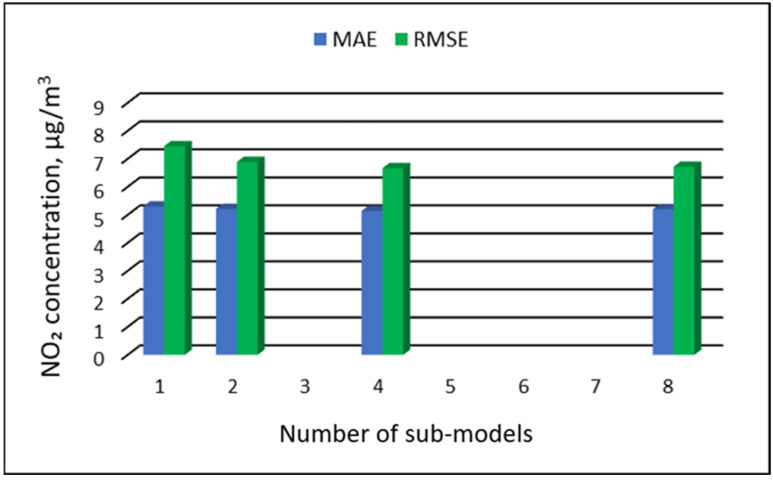
Overall MAE and RMSE values for NO_2_ concentration prediction in PVS models depending on the number of created sub-models, Zabrze.

**Figure 14 ijerph-19-16494-f014:**
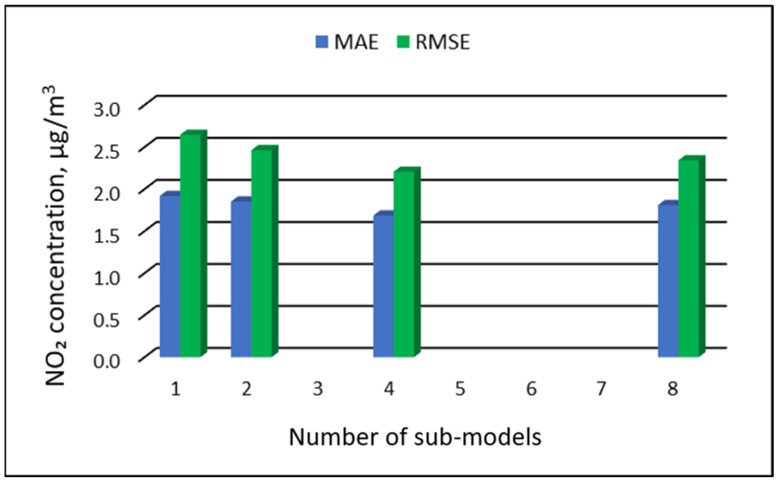
Overall MAE and RMSE values for NO_2_ concentration prediction in PVS models depending on the number of created sub-models, Złoty Potok.

**Figure 15 ijerph-19-16494-f015:**
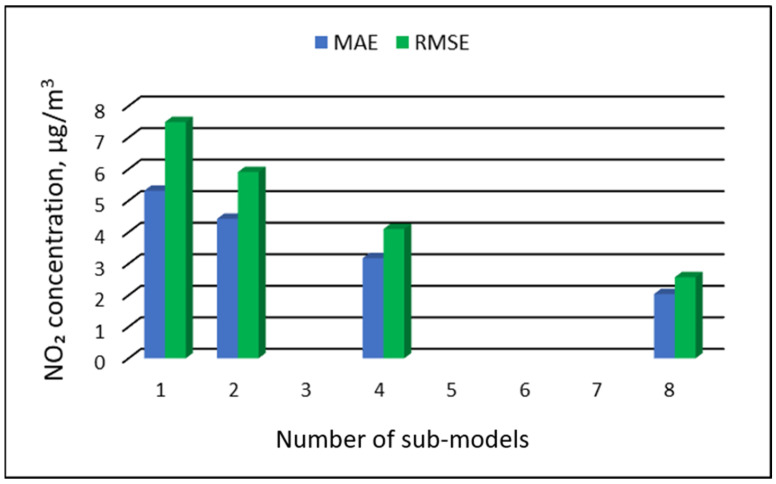
Overall MAE and RMSE values for NO_2_ concentration prediction in RVS models depending on the number of created sub-models, Zabrze.

**Figure 16 ijerph-19-16494-f016:**
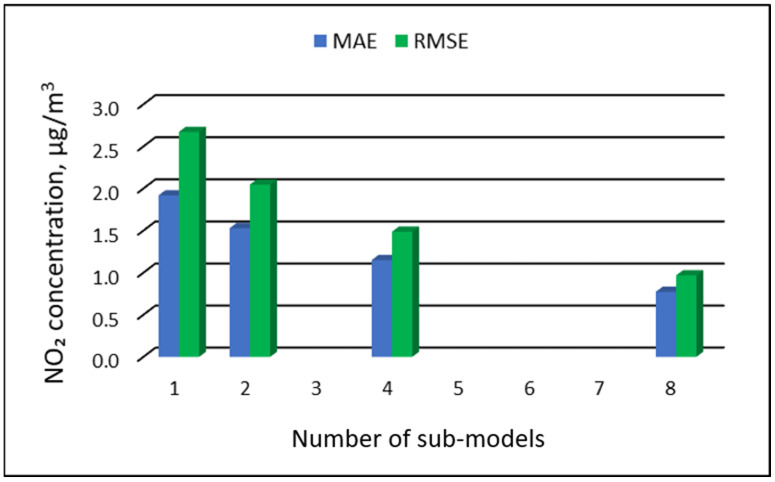
Overall MAE and RMSE values for NO_2_ concentration prediction in RVS models depending on the number of created sub-models, Złoty Potok.

**Figure 17 ijerph-19-16494-f017:**
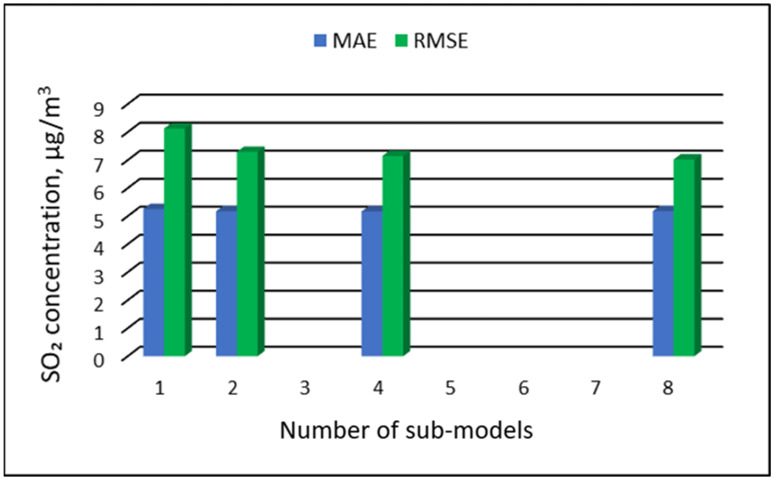
Overall MAE and RMSE values for SO_2_ concentration prediction in PVS models depending on the number of created sub-models, Zabrze.

**Figure 18 ijerph-19-16494-f018:**
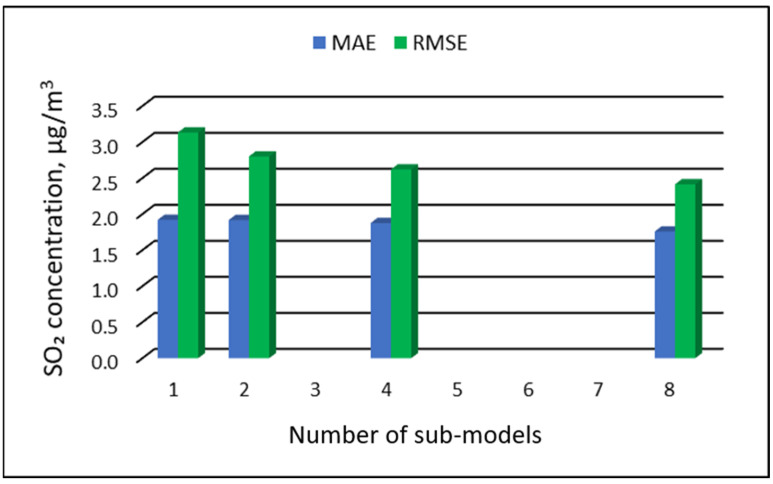
Overall MAE and RMSE values for SO_2_ concentration prediction in PVS models depending on the number of created sub-models, Złoty Potok.

**Figure 19 ijerph-19-16494-f019:**
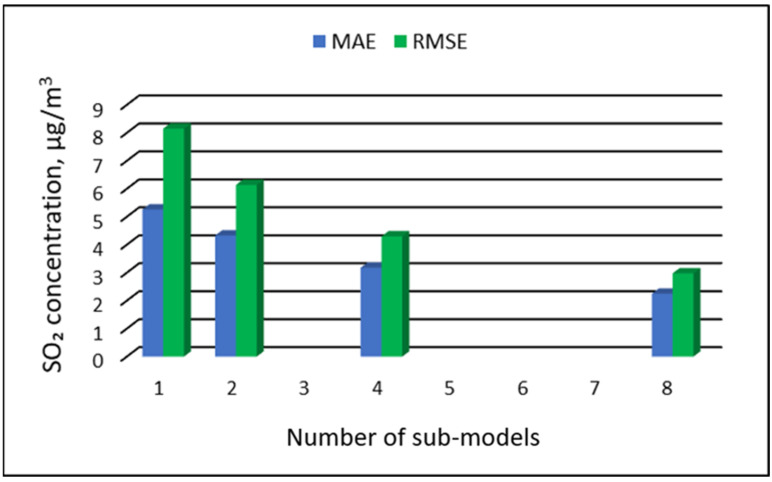
Overall MAE and RMSE values for SO_2_ concentration prediction in RVS models depending on the number of created sub-models, Zabrze.

**Figure 20 ijerph-19-16494-f020:**
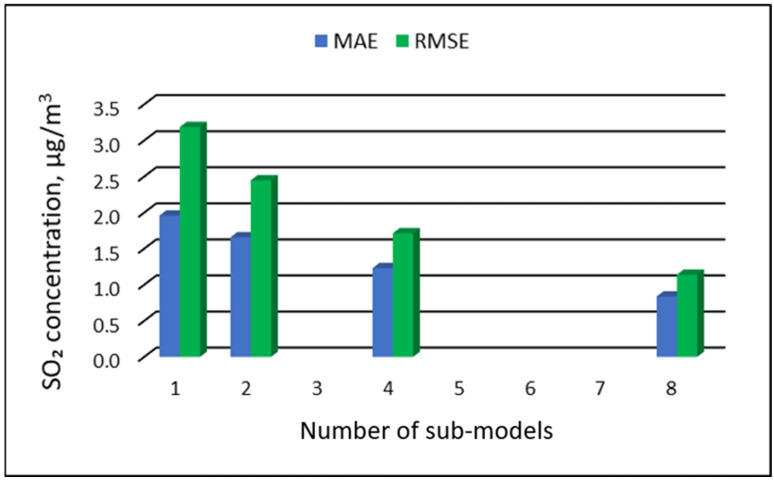
Overall MAE and RMSE values for SO_2_ concentration prediction in RVS models depending on the number of created sub-models, Złoty Potok.

**Figure 21 ijerph-19-16494-f021:**
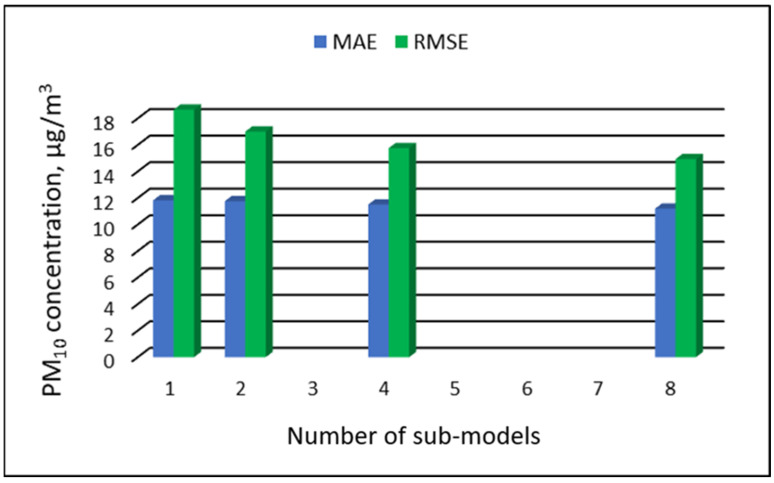
Overall MAE and RMSE values for PM_10_ concentration prediction in PVS models depending on the number of created sub-models, Zabrze.

**Figure 22 ijerph-19-16494-f022:**
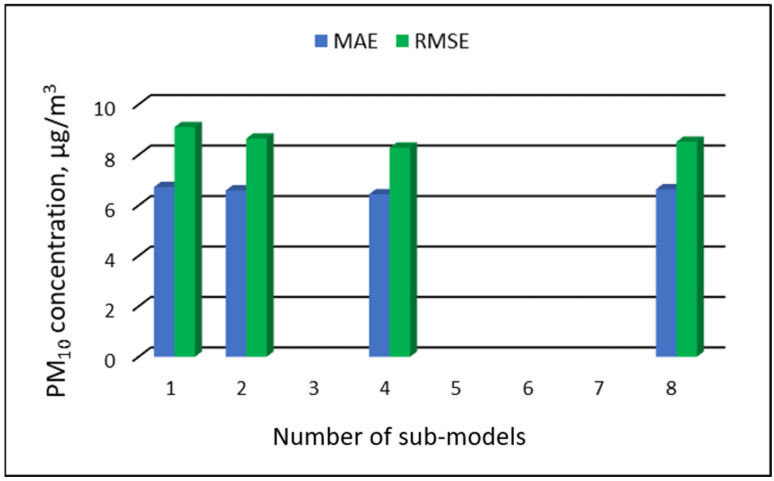
Overall MAE and RMSE values for PM_10_ concentration prediction in PVS models depending on the number of created sub-models, Złoty Potok.

**Figure 23 ijerph-19-16494-f023:**
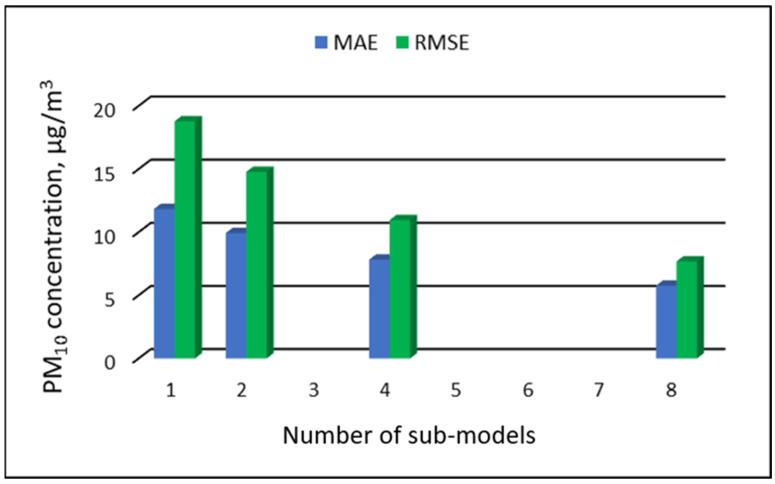
Overall MAE and RMSE values for PM_10_ concentration prediction in RVS models depending on the number of created sub-models, Zabrze.

**Figure 24 ijerph-19-16494-f024:**
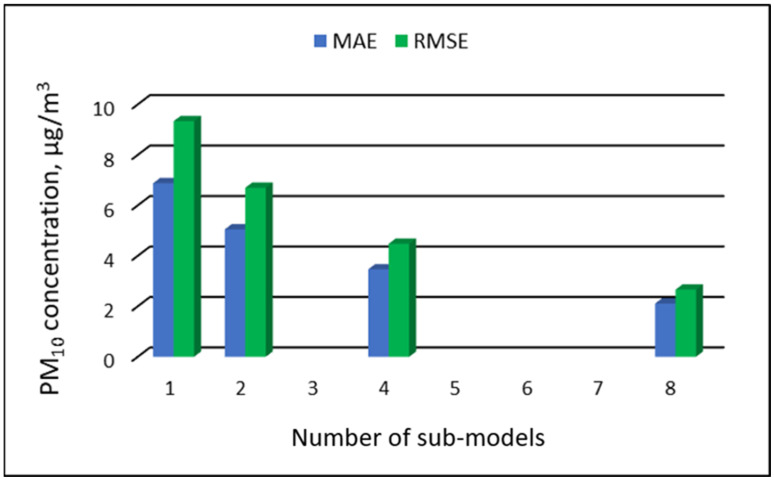
Overall MAE and RMSE values for PM_10_ concentration prediction in RVS models depending on the number of created sub-models, Złoty Potok.

**Figure 25 ijerph-19-16494-f025:**
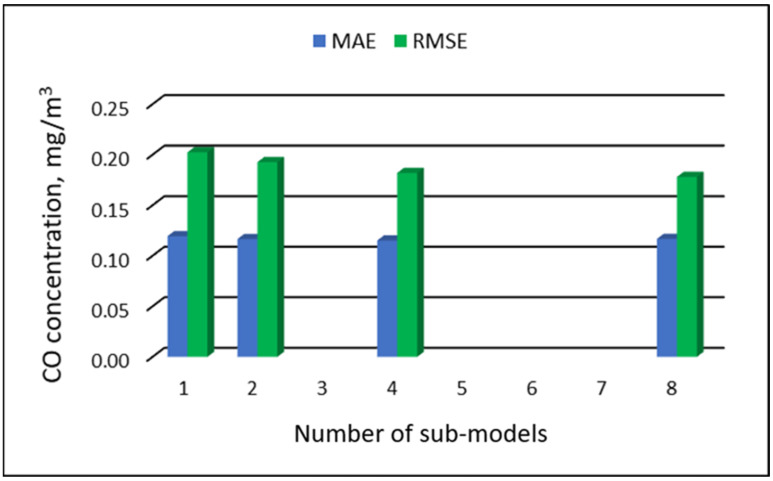
Overall MAE and RMSE values for CO concentration prediction in PVS models depending on the number of created sub-models, Zabrze.

**Figure 26 ijerph-19-16494-f026:**
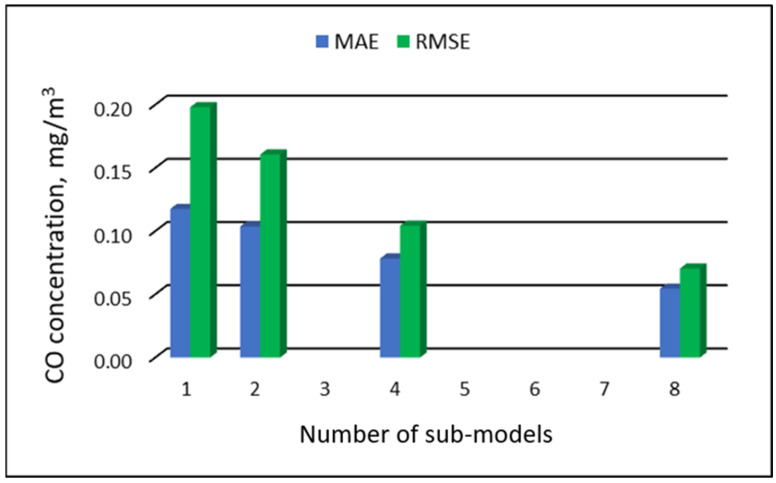
Overall MAE and RMSE values for CO concentration prediction in RVS models depending on the number of created sub-models, Zabrze.

**Table 1 ijerph-19-16494-t001:** Descriptive statistics of the set of hourly concentrations of monitored air pollutants, 2011–2016.

	Zabrze						Złoty Potok					
Descriptive Statistics	O_3_ µg/m^3^	NO µg/m^3^	NO_2_ µg/m^3^	SO_2_ µg/m^3^	PM_10_ µg/m^3^	CO mg/m^3^	O_3_ µg/m^3^	NO µg/m^3^	NO_2_ µg/m^3^	SO_2_ µg/m^3^	PM_10_ µg/m^3^	CO mg/m^3^
Minimum	0.7	0.0	1.0	0.1	0.1	0.07	1.1	0.0	0.0	0.0	1.0	-
Maximum	198.0	709.0	160.0	362.0	1187.0	9.02	212.6	50.0	82.0	116.0	298.0	-
Mean	42.3	10.2	24.4	16.8	51.0	0.60	59.5	1.1	9.6	7.4	27.8	-
Median	37.0	3.0	20.0	10.0	33.8	0.42	56.0	0.97	7.0	4.6	22.3	-
Standard deviation	32.4	24.2	16.8	19.4	61.1	0.59	31.0	1.8	7.9	8.1	20.7	-
Completeness	94.2	78.8	94.5	91.0	80.2	94.2	92.3	87.1	87.1	88.5	95.7	-

**Table 2 ijerph-19-16494-t002:** Classification of predictors for individual models at both air monitoring stations. A + symbol means the variable was treated as an input in the model. A − symbol means no measurement.

Locality	Explained Variable	Explanatory Variables (Predictors)											
		H	D	O_3_	NO	NO_2_	SO_2_	CO	PM_10_	WS	T	I	RH
Zabrze	O_3_	+	+		+	+	+	+	+	+	+	+	−
	NO	+	+	+		+	+	+	+	+	+	+	−
	NO_2_	+	+	+	+		+	+	+	+	+	+	−
	SO_2_	+	+	+	+	+		+	+	+	+	+	−
	CO	+	+	+	+	+	+		+	+	+	+	−
	PM_10_	+	+	+	+	+	+	+		+	+	+	−
Złoty Potok	O_3_	+	+		+	+	+	+	+	+	+	+	+
	NO	+	+	+		+	+	+	+	+	+	+	+
	NO_2_	+	+	+	+		+	+	+	+	+	+	+
	SO_2_	+	+	+	+	+		+	+	+	+	+	+
	CO	−	−	−	−	−	−	−	−	−	−	−	−
	PM_10_	+	+	+	+	+	+	+		+	+	+	+

**Table 3 ijerph-19-16494-t003:** Values of approximation errors of O_3_ concentrations in PVS models, Zabrze 2011–2016.

Number of Sub-Ranges	Regression Model	Sub-Ranges of O_3_ Concentrations, µg/m^3^	Number of Observations	MAE, µg/m^3^	Overall MAE, µg/m^3^	RMSE, µg/m^3^	Overall RMSE, µg/m^3^
1	PVS-1/1-ZAB	1.0–165.0	36,460	8.4	8.4	11.3	11.3
2	PVS-1/2-ZAB	1.0–37.4	18,230	6.1	8.25	8.5	10.94
	PVS-2/2-ZAB	37.4–165.0	18,230	10.4		13.4	
4	PVS-1/4-ZAB	1.0–17.7	9115	3.5	8.02	5.0	10.43
	PVS-2/4-ZAB	17.7–37.4	9115	8.45		10.8	
	PVS-3/4-ZAB	37.4–59.9	9115	9.84		12.7	
	PVS-4/4-ZAB	59.9–165.0	9115	10.3		13.2	
8	PVS-1/8-ZAB	1.0–8.6	4558	1.98	8.02	2.8	10.34
	PVS-2/8-ZAB	8.6–17.7	4558	4.81		6.3	
	PVS-3/8-ZAB	17.7–27.5	4558	7.47		9.6	
	PVS-4/8-ZAB	27.5–37.4	4558	9.4		11.9	
	PVS-5/8-ZAB	37.4–47.6	4558	9.91		12.7	
	PVS-6/8-ZAB	47.6–59.9	4558	9.98		12.9	
	PVS-7/8-ZAB	59.9–79.5	4558	9.78		12.7	
	PVS-8/8-ZAB	79.5–165.0	4558	10.81		13.7	

**Table 4 ijerph-19-16494-t004:** Values of approximation errors of O_3_ concentrations in PVS models, Złoty Potok 2011–2016.

Number of Sub-Ranges	Regression Model	Sub-Ranges of O_3_ Concentrations, µg/m^3^	Number of Observations	MAE, µg/m^3^	Overall MAE, µg/m^3^	RMSE, µg/m^3^	Overall RMSE, µg/m^3^
1	PVS-1/1-ZP	1.6–147.5	15,536	8.41	8.41	10.7	10.7
2	PVS-1/2-ZP	1.6–50.3	7768	7.66	7.91	9.9	10.09
	PVS-2/2-ZP	50.3–147.5	7768	8.16		10.3	
4	PVS-1/4-ZP	1.6–34.8	3884	6.22	7.55	8.2	9.68
	PVS-2/4-ZP	34.8–50.3	3884	8.52		10.9	
	PVS-3/4-ZP	50.3–68.7	3884	7.81		10.0	
	PVS-4/4-ZP	68.7–147.5	3884	7.64		9.6	
8	PVS-1/8-ZP	1.6–25.3	1942	5.46	7.67	7.2	9.84
	PVS-2/8-ZP	25.3–34.8	1942	8.26		10.6	
	PVS-3/8-ZP	34.8–42.9	1942	8.57		10.9	
	PVS-4/8-ZP	42.9–50.3	1942	8.69		11.3	
	PVS-5/8-ZP	50.3–58.4	1942	7.86		10.1	
	PVS-6/8-ZP	58.4–68.7	1942	7.57		9.6	
	PVS-7/8-ZP	68.7–89.9	1942	7.60		9.7	
	PVS-8/8-ZP	90.0–147.5	1942	7.36		9.3	

**Table 5 ijerph-19-16494-t005:** Values of approximation errors of O_3_ concentrations in RVS models, Zabrze 2011–2016.

Number of Sub-Ranges	Regression Model	Sub-Ranges of O_3_ Concentrations, µg/m^3^	Number of Observations	MAE, µg/m^3^	Overall MAE, µg/m^3^	RMSE, µg/m^3^	Overall RMSE, µg/m^3^
1	RVS-1/1-ZAB	0.7–198.0	36,460	8.4	8.4	11.3	11.3
2	RVS-1/2-ZAB	0.7–37.0	18,230	4.8	6.8	6.3	8.9
	RVS-2/2-ZAB	37–198.0	18,230	8.8		11.4	
4	RVS-1/4-ZAB	0.7–15.0	9115	2.2	5.1	2.8	6.4
	RVS-2/4-ZAB	15.0–37.0	9115	4.6		5.6	
	RVS-3/4-ZAB	37.0–62.0	9115	5.1		6.1	
	RVS-4/4-ZAB	62.0–198.0	9115	8.6		11.1	
8	RVS-1/8-ZAB	0.7–6.0	4558	1.0	3.3	1.2	4.0
	RVS-2/8-ZAB	6.0–15.0	4558	1.8		2.2	
	RVS-3/8-ZAB	15.0–26.0	4558	2.6		3.0	
	RVS-4/8-ZAB	26.0–37.0	4558	2.8		3.2	
	RVS-5/8-ZAB	37.0–49.0	4558	2.8		3.3	
	RVS-6/8-ZAB	49.0–62.0	4558	3.1		3.6	
	RVS-7/8-ZAB	62.0–82.0	4558	4.2		5.1	
	RVS-8/8-ZAB	82.0–198.0	4558	8.3		10.8	

**Table 6 ijerph-19-16494-t006:** Values of approximation errors of O_3_ concentrations in RVS models, Złoty Potok 2011–2016.

Number of Sub-Ranges	Regression Model	Sub-ranges of O_3_ Concentrations, µg/m^3^	Number of Observations	MAE, µg/m^3^	Overall MAE, µg/m^3^	RMSE, µg/m^3^	Overall RMSE, µg/m^3^
1	RVS-1/1-ZP	1.1–162	15,536	8.38	8.38	10.7	10.7
2	RVS-1/2-ZP	1.1–51.0	7768	5.97	6.51	7.6	8.25
	RVS-2/2-ZP	51.0–162.0	7768	7.05		8.9	
4	RVS-1/4-ZP	1.1–32.3	3884	4.06	4.67	5.1	5.84
	RVS-2/4-ZP	32.3–51.0	3884	3.85		4.7	
	RVS-3/4-ZP	51.0–72.0	3884	4.20		5.1	
	RVS-4/4-ZP	72.0–162.0	3884	6.56		8.4	
8	RVS-1/8-ZP	1.1–21.1	1942	2.98	3.06	3.6	3.74
	RVS-2/8-ZP	21.1–32.3	1942	2.48		3.0	
	RVS-3/8-ZP	32.3–42.0	1942	2.27		2.7	
	RVS-4/8-ZP	42.0–51.0	1942	2.17		2.6	
	RVS-5/8-ZP	51.0–60.4	1942	2.28		2.6	
	RVS-6/8-ZP	60.4–72.0	1942	2.56		3.1	
	RVS-7/8-ZP	72.0–90.3	1942	3.68		4.5	
	RVS-8/8-ZP	90.4–162.0	1942	6.06		7.9	

**Table 7 ijerph-19-16494-t007:** Values of approximation errors of NO concentrations in PVS models, Zabrze 2011–2016.

Number of Sub-Ranges	Regression Model	Sub-Ranges of NO Concentrations, µg/m^3^	Number of Observations	MAE, µg/m^3^	Overall MAE, µg/m^3^	RMSE, µg/m^3^	Overall RMSE, µg/m^3^
1	PVS-1/1-ZAB	0.0–709.0	36,460	3.74	3.74	8.26	8.26
2	PVS-1/2-ZAB	0.0–3.0	18,230	0.52	3.37	0.66	6.18
	PVS-2/2-ZAB	3.0–709.0	18,230	6.22		11.70	
4	PVS-1/4-ZAB	0.0–1.1	9115	0.23	2.76	0.32	4.32
	PVS-2/4-ZAB	1.1–3.0	9115	0.34		0.42	
	PVS-3/4-ZAB	3.0–8.0	9115	0.94		1.17	
	PVS-4/4-ZAB	8.0–709.0	9115	9.55		15.37	
8	PVS-1/8-ZAB	0.0–1.0	4558	0.23	2.22	0.32	3.14
	PVS-2/8-ZAB	1.0–1.1	4558	0.01		0.01	
	PVS-3/8-ZAB	1.1–2.0	4558	0.15		0.21	
	PVS-4/8-ZAB	2.0–3.0	4558	0.19		0.26	
	PVS-5/8-ZAB	3.0–4.7	4558	0.39		0.47	
	PVS-6/8-ZAB	4.7–8.0	4558	0.77		0.92	
	PVS-7/8-ZAB	8.0–19.3	4558	2.16		2.64	
	PVS-8/8-ZAB	19.4–709.0	4558	13.86		20.25	

**Table 8 ijerph-19-16494-t008:** Values of approximation errors of NO concentrations in PVS models, Złoty Potok 2011–2016.

Number of Sub-Ranges	Regression Model	Sub-Ranges of NO Concentrations, µg/m^3^	Number of Observations	MAE, µg/m^3^	Overall MAE, µg/m^3^	RMSE, µg/m^3^	Overall RMSE, µg/m^3^
1	PVS-1/1-ZP	0.1–49.8	15,536	0.459	0.459	0.789	0.789
2	PVS-1/2-ZP	0.1–0.6	7768	0.324	0.430	0.425	0.702
	PVS-2/2-ZP	0.6–49.8	7768	0.535		0.979	
4	PVS-1/4-ZP	0.1–0.4	3884	0.301	0.428	0.373	0.663
	PVS-2/4-ZP	0.4–0.6	3884	0.319		0.434	
	PVS-3/4-ZP	0.6–1.1	3884	0.331		0.501	
	PVS-4/4-ZP	1.1–49.8	3884	0.762		1.344	
8	PVS-1/8-ZP	0.1–0.3	1942	0.261	0.429	0.326	0.634
	PVS-2/8-ZP	0.3–0.4	1942	0.302		0.389	
	PVS-3/8-ZP	0.4–0.5	1942	0.311		0.398	
	PVS-4/8-ZP	0.5–0.6	1942	0.357		0.497	
	PVS-5/8-ZP	0.6–0.8	1942	0.350		0.497	
	PVS-6/8-ZP	0.8–1.1	1942	0.316		0.517	
	PVS-7/8-ZP	1.1–1.8	1942	0.386		0.586	
	PVS-8/8-ZP	1.8–49.8	1942	1.150		1.859	

**Table 9 ijerph-19-16494-t009:** Values of approximation errors of NO concentrations in RVS models, Zabrze 2011–2016.

Number of Sub-Ranges	Regression Model	Sub-Ranges of NO Concentrations, µg/m^3^	Number of Observations	MAE, µg/m^3^	Overall MAE, µg/m^3^	RMSE, µg/m^3^	Overall RMSE, µg/m^3^
1	RVS-1/1-ZAB	0.0–709.0	36,460	3.74	3.74	8.26	8.26
2	RVS-1/2-ZAB	0.0–3.0	18,230	0.52	3.37	0.66	6.18
	RVS-2/2-ZAB	3.0–709.0	18,230	6.22		11.70	
4	RVS-1/4-ZAB	0.0–1.1	9115	0.23	2.76	0.32	4.32
	RVS-2/4-ZAB	1.1–3.0	9115	0.34		0.42	
	RVS-3/4-ZAB	3.0–8.0	9115	0.94		1.17	
	RVS-4/4-ZAB	8.0–709.0	9115	9.55		15.37	
8	RVS-1/8-ZAB	0.0–1.0	4558	0.23	2.22	0.32	3.14
	RVS-2/8-ZAB	1.0–1.1	4558	0.01		0.01	
	RVS-3/8-ZAB	1.1–2.0	4558	0.15		0.21	
	RVS-4/8-ZAB	2.0–3.0	4558	0.19		0.26	
	RVS-5/8-ZAB	3.0–4.7	4558	0.39		0.47	
	RVS-6/8-ZAB	4.7–8.0	4558	0.77		0.92	
	RVS-7/8-ZAB	8.0–19.3	4558	2.16		2.64	
	RVS-8/8-ZAB	19.4–709.0	4558	13.86		20.25	

**Table 10 ijerph-19-16494-t010:** Values of approximation errors of NO concentrations in RVS models, Złoty Potok 2011–2016.

Number of Sub-Ranges	Regression Model	Sub-Ranges of NO Concentrations, µg/m^3^	Number of Observations	MAE, µg/m^3^	Overall MAE, µg/m^3^	RMSE, µg/m^3^	Overall RMSE, µg/m^3^
1	RVS-1/1-ZP	0.0–50.0	15,536	0.430	0.430	0.746	0.746
2	RVS-1/2-ZP	0.0–0.8	7768	0.184	0.310	0.234	0.571
	RVS-2/2-ZP	0.8–50.0	7768	0.436		0.908	
4	RVS-1/4-ZP	0.0–0.3	3884	0.054	0.229	0.088	0.408
	RVS-2/4-ZP	0.3–0.8	3884	0.064		0.081	
	RVS-3/4-ZP	0.8–1.0	3884	0.037		0.053	
	RVS-4/4-ZP	1.0–50.0	3884	0.760		1.411	
8	RVS-1/8-ZP	0.0–0.0	1942	-	-	-	-
	RVS-2/8-ZP	0.0–0.3	1942	0.069		0.097	
	RVS-3/8-ZP	0.3–0.6	1942	0.048		0.060	
	RVS-4/8-ZP	0.6–0.8	1942	0.041		0.049	
	RVS-5/8-ZP	0.8–1.0	1942	0.045		0.056	
	RVS-6/8-ZP	1.0–1.0	1942	-		-	
	RVS-7/8-ZP	1.0–2.0	1942	0.148		0.206	
	RVS-8/8-ZP	2.0–50.0	1942	0.991		1.752	

**Table 11 ijerph-19-16494-t011:** Values of approximation errors of NO_2_ concentrations in PVS models, Zabrze 2011–2016.

Number of Sub-Ranges	Regression Model	Sub-Ranges of NO_2_ Concentrations, µg/m^3^	Number of Observations	MAE, µg/m^3^	Overall MAE, µg/m^3^	RMSE, µg/m^3^	Overall RMSE, µg/m^3^
1	PVS-1/1-ZAB	3.6–134.4	36,460	5.3	5.3	7.4	7.4
2	PVS-1/2-ZAB	3.6–20.6	18,230	3.13	5.21	4.2	6.89
	PVS-2/2-ZAB	20.6–134.4	18,230	7.29		9.6	
4	PVS-1/4-ZAB	3.6–12.2	9115	2.29	5.14	3.0	6.66
	PVS-2/4-ZAB	12.3–20.6	9115	3.85		5.0	
	PVS-3/4-ZAB	20.6–32.8	9115	5.89		7.6	
	PVS-4/4-ZAB	32.8–134.4	9115	8.54		11.1	
8	PVS-1/8-ZAB	3.6–9.2	4558	1.80	5.20	2.4	6.72
	PVS-2/8-ZAB	9.2–12.2	4558	2.88		3.7	
	PVS-3/8-ZAB	12.3–16	4558	3.43		4.4	
	PVS-4/8-ZAB	16–20.6	4558	4.33		5.6	
	PVS-5/8-ZAB	20.6–26.2	4558	5.71		7.3	
	PVS-6/8-ZAB	26.2–32.8	4558	6.33		8.2	
	PVS-7/8-ZAB	32.8–41.7	4558	7.65		9.9	
	PVS-8/8-ZAB	41.7–134.4	4558	9.47		12.3	

**Table 12 ijerph-19-16494-t012:** Values of approximation errors of NO_2_ concentrations in PVS models, Złoty Potok 2011–2016.

Number of Sub-Ranges	Regression Model	Sub-Ranges of NO_2_ Concentrations, µg/m^3^	Number of Observations	MAE, µg/m^3^	Overall MAE, µg/m^3^	RMSE, µg/m^3^	Overall RMSE, µg/m^3^
1	PVS-1/1-ZP	1.6–52.9	15,536	1.919	1.919	2.65	2.65
2	PVS-1/2-ZP	1.6–6.7	7768	1.268	1.852	1.69	2.463
	PVS-2/2-ZP	6.7–52.9	7768	2.435		3.24	
4	PVS-1/4-ZP	1.6–4.7	3884	0.989	1.686	1.33	2.207
	PVS-2/4-ZP	4.7–6.7	3884	0.989		1.33	
	PVS-3/4-ZP	6.7–10.8	3884	1.852		2.39	
	PVS-4/4-ZP	10.8–52.9	3884	2.915		3.78	
8	PVS-1/8-ZP	1.6–3.7	1942	0.812	1.811	1.08	2.342
	PVS-2/8-ZP	3.7–4.7	1942	1.273		1.66	
	PVS-3/8-ZP	4.7–5.6	1942	1.366		1.73	
	PVS-4/8-ZP	5.6–6.7	1942	1.671		2.17	
	PVS-5/8-ZP	6.7–8.2	1942	1.786		2.26	
	PVS-6/8-ZP	8.2–10.8	1942	1.972		2.55	
	PVS-7/8-ZP	10.8–16.0	1942	2.361		3.06	
	PVS-8/8-ZP	16.1–52.9	1942	3.246		4.22	

**Table 13 ijerph-19-16494-t013:** Values of approximation errors of NO_2_ concentrations in RVS models, Zabrze 2011–2016.

Number of Sub-Ranges	Regression Model	Sub-Ranges of NO_2_ Concentrations, µg/m^3^	Number of Observations	MAE, µg/m^3^	Overall MAE, µg/m^3^	RMSE, µg/m^3^	Overall RMSE, µg/m^3^
1	RVS-1/1-ZAB	1.2–145	36,460	5.31	5.31	7.49	7.49
2	RVS-1/2-ZAB	1.2–20.0	18,230	2.37	4.42	2.97	5.91
	RVS-2/2-ZAB	20.0–145	18,230	6.47		8.84	
4	RVS-1/4-ZAB	1.2–11.5	9115	1.56	3.18	1.91	4.10
	RVS-2/4-ZAB	11.5–20.0	9115	1.82		2.19	
	RVS-3/4-ZAB	20.0–33.0	9115	2.73		3.27	
	RVS-4/4-ZAB	33.0–145.0	9115	6.61		9.03	
8	RVS-1/8-ZAB	1.2–8.0	4558	1.03	2.04	1.26	2.57
	RVS-2/8-ZAB	8.0–11.5	4558	0.86		1.00	
	RVS-3/8-ZAB	11.5–15.2	4558	0.93		1.07	
	RVS-4/8-ZAB	15.2–20.0	4558	1.12		1.32	
	RVS-5/8-ZAB	20.0–26.0	4558	1.40		1.64	
	RVS-6/8-ZAB	26.0–33.0	4558	1.68		1.97	
	RVS-7/8-ZAB	33.0–43.1	4558	2.44		2.87	
	RVS-8/8-ZAB	43.1–145.0	4558	6.84		9.44	

**Table 14 ijerph-19-16494-t014:** Values of approximation errors of NO_2_ concentrations in RVS models, Złoty Potok 2011–2016.

Number of Sub-Ranges	Regression Model	Sub-Ranges of NO_2_ Concentrations, µg/m^3^	Number of Observations	MAE, µg/m^3^	Overall MAE, µg/m^3^	RMSE, µg/m^3^	Overall RMSE, µg/m^3^
1	RVS-1/1-ZP	0.4.0–60.3	15,536	1.919	1.919	2.672	2.672
2	RVS-1/2-ZP	0.4–7.0	7768	0.894	1.529	1.110	2.047
	RVS-2/2-ZP	7.0–60.3	7768	2.164		2.985	
4	RVS-1/4-ZP	0.4–4.0	3884	0.593	1.150	0.726	1.487
	RVS-2/4-ZP	4.0–7.0	3884	0.586		0.706	
	RVS-3/4-ZP	7.0–11.1	3884	0.891		1.088	
	RVS-4/4-ZP	11.1–60.3	3884	2.529		3.426	
8	RVS-1/8-ZP	0.4–3.0	1942	0.427	0.773	0.521	0.968
	RVS-2/8-ZP	3.0–4.0	1942	0.304		0.354	
	RVS-3/8-ZP	4.0–5.4	1942	0.283		0.349	
	RVS-4/8-ZP	5.4–7.0	1942	0.332		0.408	
	RVS-5/8-ZP	7.0–8.7	1942	0.438		0.501	
	RVS-6/8-ZP	8.7–11.1	1942	0.631		0.736	
	RVS-7/8-ZP	11.1–16.6	1942	1.107		1.332	
	RVS-8/8-ZP	16.6–60.3	1942	2.661		3.544	

**Table 15 ijerph-19-16494-t015:** Values of approximation errors of SO_2_ concentrations in PVS models, Zabrze 2011–2016.

Number of Sub-Ranges	Regression Model	Sub-Ranges of SO_2_ Concentrations, µg/m^3^	Number of Observations	MAE, µg/m^3^	Overall MAE, µg/m^3^	RMSE, µg/m^3^	Overall RMSE, µg/m^3^
1	PVS-1/1-ZAB	1.3–321.9	36,460	5.25	5.25	8.1	8.1
2	PVS-1/2-ZAB	1.3–10.6	18,230	2.57	5.17	3.7	7.30
	PVS-2/2-ZAB	10.6–321.9	18,230	7.78		10.9	
4	PVS-1/4-ZAB	1.3–6.4	9115	1.81	5.17	2.6	7.15
	PVS-2/4-ZAB	6.4–10.6	9115	3.30		4.6	
	PVS-3/4-ZAB	10.6–21.3	9115	5.88		8.1	
	PVS-4/4-ZAB	21.3–321.9	9115	9.69		13.3	
8	PVS-1/8-ZAB	1.3–4.9	4558	1.49	5.18	2.2	7.02
	PVS-2/8-ZAB	4.9–6.4	4558	2.19		3.0	
	PVS-3/8-ZAB	6.4–8.2	4558	2.85		3.9	
	PVS-4/8-ZAB	8.2–10.6	4558	3.85		5.3	
	PVS-5/8-ZAB	10.6–14.4	4558	5.46		7.5	
	PVS-6/8-ZAB	14.4–21.3	4558	6.69		9.1	
	PVS-7/8-ZAB	21.3–34.0	4558	7.08		9.3	
	PVS-8/8-ZAB	34.0–321.9	4558	11.82		15.9	

**Table 16 ijerph-19-16494-t016:** Values of approximation errors of SO_2_ concentrations in PVS models, Złoty Potok 2011–2016.

Number of Sub-Ranges	Regression Model	Sub-Ranges of SO_2_ Concentrations, µg/m^3^	Number of Observations	MAE, µg/m^3^	Overall MAE, µg/m^3^	RMSE, µg/m^3^	Overall RMSE, µg/m^3^
1	PVS-1/1-ZP	0.8–74.2	15,536	1.920	1.920	3.133	3.133
2	PVS-1/2-ZP	0.8–4.0	7768	0.935	1.917	1.302	2.800
	PVS-2/2-ZP	4.0–74.2	7768	2.899		4.298	
4	PVS-1/4-ZP	0.8–2.8	3884	0.823	1.876	1.099	2.620
	PVS-2/4-ZP	2.8–4.0	3884	1.135		1.597	
	PVS-3/4-ZP	4.0–7.2	3884	1.710		2.395	
	PVS-4/4-ZP	7.2–74.2	3884	3.836		5.389	
8	PVS-1/8-ZP	0.8–2.2	1942	0.699	1.759	0.899	2.413
	PVS-2/8-ZP	2.2–2.8	1942	0.855		1.131	
	PVS-3/8-ZP	2.8–3.3	1942	1.008		1.379	
	PVS-4/8-ZP	3.3–4.0	1942	1.246		1.778	
	PVS-5/8-ZP	4.0–5.1	1942	0.057		0.080	
	PVS-6/8-ZP	5.1–7.2	1942	2.024		2.813	
	PVS-7/8-ZP	7.2–11.8	1942	3.108		4.545	
	PVS-8/8-ZP	11.8–74.2	1942	5.072		6.681	

**Table 17 ijerph-19-16494-t017:** Values of approximation errors of SO_2_ concentrations in RVS models, Zabrze 2011–2016.

Number of Sub-Ranges	Regression Model	Sub-Ranges of SO_2_ Concentrations, µg/m^3^	Number of Observations	MAE, µg/m^3^	Overall MAE, µg/m^3^	RMSE, µg/m^3^	Overall RMSE, µg/m^3^
1	RVS-1/1-ZAB	0.1–362	36,460	5.26	5.26	8.15	8.15
2	RVS-1/2-ZAB	0.1–10.0	18,230	1.58	4.33	1.95	6.14
	RVS-2/2-ZAB	10.0–362.0	18,230	7.08		10.33	
4	RVS-1/4-ZAB	0.1–5.0	9115	0.85	3.18	1.03	4.30
	RVS-2/4-ZAB	5.0–10	9115	1.11		1.32	
	RVS-3/4-ZAB	10.0–22.3	9115	2.66		3.21	
	RVS-4/4-ZAB	22.4–362.0	9115	8.11		11.64	
8	RVS-1/8-ZAB	0.1–3.2	4558	0.55	2.25	0.64	2.98
	RVS-2/8-ZAB	3.2–5.0	4558	0.41		0.48	
	RVS-3/8-ZAB	5.0–7.0	4558	0.48		0.56	
	RVS-4/8-ZAB	7.0–10	4558	0.73		0.85	
	RVS-5/8-ZAB	10.0–14.9	4558	1.16		1.36	
	RVS-6/8-ZAB	14.9–22.3	4558	1.82		2.14	
	RVS-7/8-ZAB	22.4–36.0	4558	3.02		3.58	
	RVS-8/8-ZAB	36.0–362.0	4558	9.84		14.21	

**Table 18 ijerph-19-16494-t018:** Values of approximation errors of SO_2_ concentrations in RVS models, Złoty Potok 2011–2016.

Number of Sub-Ranges	Regression Model	Sub-Ranges of SO_2_ Concentrations, µg/m^3^	Number of Observations	MAE, µg/m^3^	Overall MAE, µg/m^3^	RMSE, µg/m^3^	Overall RMSE, µg/m^3^
1	RVS-1/1-ZP	0.0–85.0	15,536	1.96	1.96	3.186	3.186
2	RVS-1/2-ZP	0.0–4.0	7768	0.63	1.66	0.780	2.45
	RVS-2/2-ZP	4.0–85.0	7768	2.68		4.116	
4	RVS-1/4-ZP	0.0–2.1	3884	0.38	1.23	0.455	1.71
	RVS-2/4-ZP	2.1–4.0	3884	0.36		0.461	
	RVS-3/4-ZP	4.0–7.4	3884	0.73		0.891	
	RVS-4/4-ZP	7.4–85.0	3884	3.45		5.048	
8	RVS-1/8-ZP	0.0–1.8	1942	0.26	0.84	0.333	1.14
	RVS-2/8-ZP	1.8–2.1	1942	0.04		0.063	
	RVS-3/8-ZP	2.1–3.0	1942	0.19		0.245	
	RVS-4/8-ZP	3.0–4.0	1942	0.23		0.289	
	RVS-5/8-ZP	4.0–5.0	1942	0.34		0.385	
	RVS-6/8-ZP	5.0–7.4	1942	0.58		0.683	
	RVS-7/8-ZP	7.4–12.0	1942	1.01		1.216	
	RVS-8/8-ZP	12.0–85.0	1942	4.05		5.909	

**Table 19 ijerph-19-16494-t019:** Values of approximation errors of PM_10_ concentrations in PVS models, Zabrze 2011–2016.

Number of Sub-Ranges	Regression Model	Sub-Ranges of PM_10_ Concentrations, µg/m^3^	Number of Observations	MAE, µg/m^3^	Overall MAE, µg/m^3^	RMSE, µg/m^3^	Overall RMSE, µg/m^3^
1	PVS-1/1-ZAB	8.2–980.5	36,460	11.83	11.83	18.7	18.7
2	PVS-1/2-ZAB	8.2–33.1	18,230	6.53	11.76	8.79	17.00
	PVS-2/2-ZAB	33.1–980.5	18,230	16.99		25.22	
4	PVS-1/4-ZAB	8.2–22.1	9115	5.52	11.51	7.6	15.76
	PVS-2/4-ZAB	22.1–33.1	9115	7.42		9.7	
	PVS-3/4-ZAB	33.1–55.6	9115	10.01		13.2	
	PVS-4/4-ZAB	55.6–980.5	9115	23.07		32.6	
8	PVS-1/8-ZAB	8.2–17.9	4558	4.47	11.20	5.9	14.93
	PVS-2/8-ZAB	17.9–22.1	4558	6.39		8.8	
	PVS-3/8-ZAB	22.1–26.9	4558	6.72		8.9	
	PVS-4/8-ZAB	26.9–33.1	4558	8.24		10.7	
	PVS-5/8-ZAB	33.1–41.2	4558	9.18		12.1	
	PVS-6/8-ZAB	41.2–55.6	4558	11.11		14.4	
	PVS-7/8-ZAB	55.6–90.1	4558	14.67		19.0	
	PVS-8/8-ZAB	90.2–980.5	4558	28.79		39.7	

**Table 20 ijerph-19-16494-t020:** Values of approximation errors of PM_10_ concentrations in PVS models, Złoty Potok 2011–2016.

Number of Sub-Ranges	Regression Model	Sub-Ranges of PM_10_ Concentrations, µg/m^3^	Number of Observations	MAE, µg/m^3^	Overall MAE, µg/m^3^	RMSE, µg/m^3^	Overall RMSE, µg/m^3^
1	PVS-1/1-ZP	5.1–120.4	15,536	6.726	6.726	9.10	9.10
2	PVS-1/2-ZP	5.1–22.7	7768	4.638	6.589	6.11	8.646
	PVS-2/2-ZP	22.7–120.4	7768	8.541		11.18	
4	PVS-1/4-ZP	5.1–18.0	3884	3.964	6.442	5.12	8.285
	PVS-2/4-ZP	18.0–22.7	3884	5.230		6.76	
	PVS-3/4-ZP	22.7–30.9	3884	6.602		8.57	
	PVS-4/4-ZP	30.9–120.4	3884	9.972		12.70	
8	PVS-1/8-ZP	5.1–15.7	1942	3.406	6.639	4.39	8.519
	PVS-2/8-ZP	15.7–18.0	1942	4.634		6.01	
	PVS-3/8-ZP	18.0–20.2	1942	4.963		6.43	
	PVS-4/8-ZP	20.2–22.7	1942	5.613		7.20	
	PVS-5/8-ZP	22.7–25.9	1942	6.751		8.74	
	PVS-6/8-ZP	25.9–30.9	1942	7.202		9.28	
	PVS-7/8-ZP	30.9–40.5	1942	8.027		10.23	
	PVS-8/8-ZP	40.5–120.4	1942	12.516		15.88	

**Table 21 ijerph-19-16494-t021:** Values of approximation errors of PM_10_ concentrations in RVS models, Zabrze 2011–2016.

Number of Sub-Ranges	Regression Model	Sub-Ranges of PM_10_ Concentrations, µg/m^3^	Number of Observations	MAE, µg/m^3^	Overall MAE, µg/m^3^	RMSE, µg/m^3^	Overall RMSE, µg/m^3^
1	RVS-1/1-ZAB	0.1–1145.0	36,460	11.84	11.84	18.8	18.8
2	RVS-1/2-ZAB	0.1–33.0	18,230	4.66	9.93	5.8	14.77
	RVS-2/2-ZAB	33.0–1145.0	18,230	15.20		23.8	
4	RVS-1/4-ZAB	0.1–20.0	9115	3.09	7.83	3.8	10.96
	RVS-2/4-ZAB	20.0–33.0	9115	2.97		3.5	
	RVS-3/4-ZAB	33.0–57.1	9115	5.02		6.1	
	RVS-4/4-ZAB	57.1–1145.0	9115	20.23		30.4	
8	RVS-1/8-ZAB	0.1–14.0	4558	2.26	5.76	2.8	7.68
	RVS-2/8-ZAB	14.0–20.0	4558	1.49		1.7	
	RVS-3/8-ZAB	20.0–26.0	4558	1.51		1.8	
	RVS-4/8-ZAB	26.0–33.0	4558	1.74		2.0	
	RVS-5/8-ZAB	33.0–42.5	4558	2.30		2.7	
	RVS-6/8-ZAB	42.5–57.1	4558	3.51		4.1	
	RVS-7/8-ZAB	57.1–91.1	4558	6.99		8.4	
	RVS-8/8-ZAB	91.1–1145.0	4558	26.32		38.0	

**Table 22 ijerph-19-16494-t022:** Values of approximation errors of PM_10_ concentrations in RVS models, Złoty Potok 2011–2016.

Number of Sub-Ranges	Regression Model	Sub-Ranges of PM_10_ Concentrations, µg/m^3^	Number of Observations	MAE, µg/m^3^	Overall MAE, µg/m^3^	RMSE, µg/m^3^	Overall RMSE, µg/m^3^
1	RVS-1/1-ZP	3.0–138.0	15,536	6.88	6.88	9.34	9.34
2	RVS-1/2-ZP	3.0–22.8	7768	2.99	5.05	3.64	6.70
	RVS-2/2-ZP	22.8–138.0	7768	7.11		9.75	
4	RVS-1/4-ZP	3.0–16.0	3884	1.88	3.46	2.31	4.48
	RVS-2/4-ZP	16.0–22.8	3884	1.62		1.90	
	RVS-3/4-ZP	22.8–32.9	3884	2.29		2.71	
	RVS-4/4-ZP	32.9–138.0	3884	8.07		11.00	
8	RVS-1/8-ZP	3.0–12.1	1942	1.43	2.11	1.76	2.66
	RVS-2/8-ZP	12.1–16.0	1942	0.84		0.99	
	RVS-3/8-ZP	16.0–19.0	1942	0.84		0.97	
	RVS-4/8-ZP	19.0–22.8	1942	0.89		1.03	
	RVS-5/8-ZP	22.8–27.0	1942	1.04		1.21	
	RVS-6/8-ZP	27.0–32.9	1942	1.46		1.69	
	RVS-7/8-ZP	32.9–43.0	1942	2.50		2.94	
	RVS-8/8-ZP	43.0–138.0	1942	7.88		10.71	

**Table 23 ijerph-19-16494-t023:** Values of approximation errors of CO concentrations in PVS models, Zabrze 2011–2016.

Number of Sub-Ranges	Regression Model	Sub-Ranges of CO Concentrations, µg/m^3^	Number of Observations	MAE, µg/m^3^	Overall MAE, µg/m^3^	RMSE, µg/m^3^	Overall RMSE, µg/m^3^
1	PVS-1/1-ZAB	0.16–8.28	36,460	0.119	0.119	0.202	0.202
2	PVS-1/2-ZAB	0.16–0.42	18,230	0.071	0.117	0.155	0.193
	PVS-2/2-ZAB	0.42–8.28	18,230	0.162		0.230	
4	PVS-1/4-ZAB	0.16–0.30	9115	0.066	0.115	0.196	0.182
	PVS-2/4-ZAB	0.30–0.42	9115	0.074		0.098	
	PVS-3/4-ZAB	0.42–0.69	9115	0.111		0.149	
	PVS-4/4-ZAB	0.69–8.28	9115	0.210		0.285	
8	PVS-1/8-ZAB	0.16–0.26	4558	0.072	0.117	0.198	0.178
	PVS-2/8-ZAB	0.26–0.30	4558	0.069		0.177	
	PVS-3/8-ZAB	0.30–0.35	4558	0.065		0.085	
	PVS-4/8-ZAB	0.35–0.42	4558	0.079		0.106	
	PVS-5/8-ZAB	0.42–0.52	4558	0.099		0.132	
	PVS-6/8-ZAB	0.52–0.69	4558	0.121		0.158	
	PVS-7/8-ZAB	0.69–1.03	4558	0.153		0.203	
	PVS-8/8-ZAB	1.03–8.28	4558	0.275		0.365	

**Table 24 ijerph-19-16494-t024:** Values of approximation errors of CO concentrations in RVS models, Zabrze 2011–2016.

Number of Sub-Ranges	Regression Model	Sub-Ranges of CO Concentrations, µg/m^3^	Number of Observations	MAE, µg/m^3^	Overall MAE, µg/m^3^	RMSE, µg/m^3^	Overall RMSE, µg/m^3^
1	RVS-1/1-ZAB	0.1–9.0	36,460	0.118	0.118	0.198	0.198
2	RVS-1/2-ZAB	0.1–0.4	18,230	0.047	0.104	0.059	0.161
	RVS-2/2-ZAB	0.4–9.0	18,230	0.160		0.262	
4	RVS-1/4-ZAB	0.1–0.3	9115	0.034	0.078	0.042	0.104
	RVS-2/4-ZAB	0.3–0.4	9115	0.030		0.035	
	RVS-3/4-ZAB	0.4–0.7	9115	0.055		0.066	
	RVS-4/4-ZAB	0.7–9.0	9115	0.196		0.274	
8	RVS-1/8-ZAB	0.1–0.2	4558	0.027	0.055	0.033	0.070
	RVS-2/8-ZAB	0.2–0.3	4558	0.014		0.016	
	RVS-3/8-ZAB	0.3–0.3	4558	0.014		0.017	
	RVS-4/8-ZAB	0.3–0.4	4558	0.018		0.021	
	RVS-5/8-ZAB	0.4–0.5	4558	0.025		0.030	
	RVS-6/8-ZAB	0.5–0.7	4558	0.038		0.045	
	RVS-7/8-ZAB	0.7–1.1	4558	0.077		0.093	
	RVS-8/8-ZAB	1.1–9.0	4558	0.223		0.309	

**Table 25 ijerph-19-16494-t025:** The percentage changes in the overall values of MAE and RMSE obtained by modeling the concentrations in sub-ranges, in relation to the MAE and RMSE values of the corresponding full-range models.

Air Pollutant	Number of Sub-Models	Zabrze				Zloty Potok			
		PVS Sub-Models		RVS Sub-Models		PVS Sub-Models		RVS Sub-Models	
		ΔMAE %	ΔRMSE %	ΔMAE %	ΔRMSE %	ΔMAE %	ΔRMSE %	ΔMAE %	ΔRMSE %
O_3_	2	−2.1	−2.9	−19.2	−21.3	−5.9	−5.6	−22.3	−23.0
	4	−4.9	−7.4	−39.5	−43.4	−10.2	−9.5	−44.3	−45.5
	8	−4.9	−8.2	−60.7	−64.1	−8.7	−8.0	−63.5	−65.1
NO	2	−4.6	−22.9	−9.9	−25.2	−6.4	−11.0	−27.9	−23.5
	4	−7.7	−36.9	−26.1	−47.7	−6.7	−16.0	−46.8	−45.3
	8	−8.3	−41.9	−40.6	−62.0	−6.5	−19.7	-	-
NO_2_	2	−1.7	−7.5	−16.8	−21.2	−3.5	−7.0	−20.3	−23.4
	4	−3.0	−10.5	−40.2	−45.3	−12.1	−16.6	−40.1	−44.4
	8	−1.9	−9.8	−61.6	−65.7	−5.6	−11.5	−59.7	−63.8
SO_2_	2	−1.5	−10.2	−17.7	−24.7	−0.1	−10.6	−15.4	−23.2
	4	−1.6	−12.0	−39.5	−47.3	−2.3	−16.4	−37.2	−46.2
	8	−1.4	−13.6	−57.2	−63.5	−8.4	−23.0	−57.2	−64.2
PM_10_	2	−0.5	−8.9	−16.1	−21.3	−2.0	−5.0	−26.6	−28.2
	4	−2.7	−15.5	−33.9	−41.7	−4.2	−9.0	−49.6	−52.0
	8	−5.3	−20.0	−51.3	−59.1	−1.3	−6.4	−69.3	−71.5
CO	2	−2.4	−4.8	−11.8	−18.9	-	-	-	-
	4	−3.5	−10.1	−33.4	−47.3	-	-	-	-
	8	−2.2	−12.1	−53.6	−64.5	-	-	-	-

## Data Availability

Not applicable.

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
