# Peer review of "Air Quality Modeling with the Use of Regression Neural Networks"

_ijerph, 2022, doi:10.3390/ijerph192416494_

Round 1

Reviewer 1 Report (Previous Reviewer 1)

Please see attached review

Author Response

Response to Reviewer 1

The authors are grateful for any comments. Taking into account the valuable comments of the Reviewers made it possible to improve the quality of the submitted article. Most of the Reviewers' comments were included in the new version of the article.

Below are the details of the manuscript revisions and responses, point by point.

  1. The authors should state that PM2.5 measurements are not available (maybe near L115). The statement in the introduction “the main air pollutants include…” (L46) would be better with PM2.5 included rather than PM10.

Response:

We cannot write that PM2.5 measurements were not available, because at one of the stations (Zloty Potok) this measurement was started in 2013. However, we were warned that the reliability of these measurements is high. In addition, the measurements covered only one of the stations and only selected years from 2011-2016. We chose PM10 because these measurements were available at both stations and throughout the measurement period.

The authors do not fully agree with the Reviewer's suggestion that PM2.5, rather than PM10, should be considered the main air pollutant. Of course, PM2.5 is very important from the point of view of health aspects but for several decades, in conventional air quality monitoring, the measurement of PM10 concentration was the only commonly used measurement of particulate matter. In recent years, the measurement of PM2.5 concentration and even finer PM fractions has been introduced, but the measurement of PM10 concentration is still the most common. The authors of the article consider the measurement of PM10 concentration to be basic, and the measurement of PM2.5 concentration to be supplementary.

As You know, the PM2.5 fraction is part of the PM10 fraction. In Poland, the share of the PM2.5 fraction in PM10 is very high, usually ranging from 70-80% on average. The PM10 measurement cannot formally replace the PM2.5 measurement, but it contains information about the expected levels of PM2.5 concentrations.

We would like to point out that the essence of the article is not to assess concentration levels and their impact on health, but to assess the possibility of reducing the modeling error. We are convinced that in this respect the conclusions resulting from the modeling of PM10 concentrations would be consistent with the conclusions resulting from the analysis of PM2.5.

  1. L92-97: I disagree with the authors that all of this methodological information belongs in the introduction. Look at this last sentence of the paragraph. It is going beyond the purpose and scope of the paper and unnecessarily into the method. Please remove (or move to Sect. 2) everything starting with “Multilayer …” until the end of the paragraph.

Response:

This passage has been moved to subsection 3.2. Regression Models.

  1. L156: More readers would be able to under the method if the authors briefly explain ‘epochs’.

Response:

Appropriate additions have been made to the text.

Lines 169-172.

  1. L159: The authors responded to “5 models” in their reply, but the paper needs to be improved in this regard. The reader has no idea why different models give different results and, at this point, do not have a clue how the most accurate one is determined. This section requires more information. Does it make more sense to combine Sect. 2.2 and 2.3?

Response:

Appropriate changes and additions have been made to the text.

Lines 175-184.

We felt it made more sense to keep Subsections 2.2 and 2.3 separate. In a practical sense, these are 2 separate problems. In our opinion, such text is easier to read.

Reviewer 2 Report (Previous Reviewer 2)

After substantial revisions and putting more references, the paper is written well, and the text is clear and easy to read.
The conclusions are consistent with the evidence and arguments presented.

Author Response

The authors are grateful for any comments. Taking into account the valuable comments of the Reviewers made it possible to improve the quality of the submitted article. Most of the Reviewers' comments were included in the new version of the article.

Reviewer 3 Report (New Reviewer)

This paper has verified through a large number of experiments that the prediction model created by using different pollutant concentration sub ranges can improve the accuracy of pollutant concentration in the neural regression model. The perspective is innovative. However it would be valueable with the necessary revision.

1.There are a large number of irregular red marks in the text, and it is hoped that the format can be unified.

2.In Section 2.1,if the dataset is public,please set a link to get it.If not,please explain.

3.In Section2,the table 2 makes reader difficult to understand.It would be comprehensible if you do a simple explain to it.

4.We have noticed that the error of last group in every subgroups(2 subgroups,4 subgroups,8 subgroups) in the experiment is always very large.But it seems to be no explanation in this paper.It would be better if a reasonable explanation could be added.

5.In the conclusion,it would be better if the author can explain the shortcomings and limitations of this paper.

Author Response

Response to Reviewer 3

The authors are grateful for any comments. Taking into account the valuable comments of the Reviewers made it possible to improve the quality of the submitted article. Most of the Reviewers' comments were included in the new version of the article.

Below are the details of the manuscript revisions and responses, point by point. Our responses were marked blue.

  1. There are a large number of irregular red marks in the text, and it is hoped that the format can be unified.

Response:

The red marks in the text are corrections and additions introduced earlier. The editors require that the changes be introduced using the "track changes" function. In the latest version of the article, only the latest corrections and changes are marked.

  1. In Section 2.1, if the dataset is public, please set a link to get it. If not, please explain.

Response:

The analyzed data sets are not publicly available. We received them on an individual request from the Provincial Inspectorate for Environmental Protection. Time series of concentrations are available on the website of the Chief Inspectorate of Environmental Protection: https://powietrze.gios.gov.pl/pjp/archives. This is only part of the data. Meteorological data is not available online.

Appropriate additions have been made to the text. Lines 113-118.

  1. In Section2, the table 2 makes reader difficult to understand. It would be comprehensible if you do a simple explain to it.

Appropriate changes and additions have been made to the text.

Lines 145-150.

  1. We have noticed that the error of last group in every subgroups (2 subgroups, 4 subgroups, 8 subgroups) in the experiment is always very large. But it seems to be no explanation in this paper. It would be better if a reasonable explanation could be added.

Appropriate changes and additions have been made to the text.

Lines 543-548.

  1. In the conclusion, it would be better if the author can explain the shortcomings and limitations of this paper.

Appropriate additions have been made to the text.

Lines 587-595.

Round 2

Reviewer 3 Report (New Reviewer)

The authors have addressed all my concerns.

This manuscript is a resubmission of an earlier submission. The following is a list of the peer review reports and author responses from that submission.

Round 1

Reviewer 1 Report

 The paper is rejected because of a lack of methodological detail. 

Review of Hoffman et al. (submitted to Int. J. Env. Res. Public. Health)

This is yet another neural network paper which does not contain sufficient detail on the methods used. The introduction is very good. I reject the paper because the methods are not adequately described. Also, the authors should comment on whether PM2.5 measurements are available. Presumably they are not, otherwise they should have been used since PM2.5 is much more closely related to poor air quality and human health problems. It is not clear why PVS model errors using 8 sub-ranges are sometimes worse than for 4 sub-ranges. No insight is given in the discussion (Sect. 4).

Specific comments

L35: The authors should know, and maybe they already do, that certain pollutants like SO2 and NO2 have a short lifetime in the boundary layer and do not spread globally.

L36: “in Poland” -> “, in Poland, ”

L37: Delete “are”

L54: “EU” -> “the EU”

L89-97: These lines should not be in the introduction. They briefly describe the method. The method needs to be described in greater detail than this and I suggest that the authors simply delete these sentences since they are essentially repeated in section 2.

L110: Potok -> Potok,                                   (a comma should follow all leading prepositional phrases)

L130: HR -> RH                                               (more conventional)

L141: BFGS is not defined and a literature reference is not given.

L142: What is an epoch in this context? Is 300 epochs basically 300 hours?

L145: The created models are not presented later. The reader will not understand why the 5 models are different. Is it because different time periods are used for each model?

L180: It is completely unclear how the best regression model is obtained when real data are unavailable.   

L206: How can there be “actual values” when L176 states that the real concentrations are not known?

L493: How is the preliminary prediction done? 

Reviewer 2 Report

Manuscript Number; IJERPH_2017979

Title; Air quality modeling with the use of regression neural networks

Evaluation; Minor Revision.

1.    The data was recorded in Zabrze and ZÅ‚oty Potok in the years 2011–2016,

           Do they address the main question posed?

           Can the data be considered updated now? In my personal opinion, yes because the atmospheric pollution in those areas is not improved. Consequently, the data could be only underrated. The authors should write something about the 5 year’s time period between the data acquisition and the presented paper (2017 - 2022).

The paper is featured solid data, but all of them are referred to 2011-2016 time period. So, even if the environmental situation shouldn't been improved in Poland until today, some consideration about it should be reported in the "Conclusions" paragraph.    

Reviewer 3 Report

In summary, this study has a small contribution to the air pollution modelling area.

First of all, to be straightforward, this study is merely a missing gap-filling method paper. Filling the missing gap at a  monitoring station on an hourly level is never a significant issue in this area. A 10% of missing data is often seen in most of the EU, USA and China monitoring stations for various purposes such as monitor malfunction.

Legislations pay small attention to hourly concentrations, instead, annual concentration levels were referred for legislation purposes (https://www.who.int/news/item/22-09-2021-new-who-global-air-quality-guidelines-aim-to-save-millions-of-lives-from-air-pollution).  There is plenty of method paper in the area of computer science of such kind (https://link.springer.com/article/10.1007/s10462-019-09709-4). In terms of health assessment, current literature is using annual, monthly or weekly exposure depending on the health outcomes.  Missing 10% of data would not impact the assessment significantly.

Secondly, the method section of this paper is unclear. I need more information on summary statistics of the studied pollutants. How did the author treat the outliers?

Also, the neural network structure is unclear. You need to present more information on the neural network layers. Finally, the cited article is outdated. For example, Artificial neural networks (the multilayer perceptron)—a review of applications in the atmospheric sciences is an article published in 1998, which is 24 year ago.